# Adaptable Safe Policy Learning from Multi-task Data with Constraint Prioritized Decision Transformer

**Ruiqi Xue[1,2] , Ziqian Zhang[1,2], Lihe Li[1,2], Cong Guan[1,2], Lei Yuan[1,2,3]\* Yang Yu[1,2,3]\***

[1] National Key Laboratory for Novel Software Technology, Nanjing University
[2] School of Artificial Intelligence, Nanjing University
[3] Polixir Technologies
Nanjing, China
`{xuerq, zhangzq, lilh, guanc, yuanl}@lamda.nju.edu.cn`
`yuy@nju.edu.cn`

## Abstract

Learning safe reinforcement learning (RL) policies from offline multi-task datasets without direct environmental interaction is crucial for efficient and reliable deployment of RL agents. Benefiting from their scalability and strong in-context learning capabilities, recent approaches attempt to utilize Decision Transformer (DT) architectures for offline safe RL, demonstrating promising adaptability across varying safety budgets. However, these methods primarily focus on single-constraint scenarios and struggle with diverse constraint configurations across multiple tasks. Additionally, their reliance on heuristically defined Return-To-Go (RTG) inputs limits flexibility and reduces learning efficiency, particularly in complex multi-task scenarios. To address these limitations, we propose CoPDT, a novel DT-based framework designed to enhance adaptability to diverse constraints (i.e., cost functions) and varying budgets. Specifically, CoPDT introduces a constraint prioritized prompt encoder, which leverages sparse binary cost signals to accurately identify constraints, and a constraint prioritized Return-To-Go (CPRTG) token mechanism, which dynamically generates RTGs based on identified constraints and corresponding safety budgets. Extensive experiments on the OSRL benchmark demonstrate that CoPDT achieves superior efficiency and significantly enhanced safety compliance across diverse multi-task scenarios, surpassing state-of-the-art DT-based methods by satisfying safety constraints in more than twice as many tasks.

## 1 Introduction

Safe reinforcement learning (RL), focusing on deriving policies that explicitly satisfy predefined safety constraints, has garnered significant attention due to its applicability in critical domains such as autonomous driving (Zhang et al., 2021), robotic control (Brunke et al., 2022), and aligning large language models with human values (Dai et al., 2024). However, conventional online RL methods rely heavily on inherently risky trial-and-error interactions, limiting their practical deployment in safety-critical environments (Xu et al., 2022b). To address this limitation, offline safe RL, aiming to learn safe policies purely from previously collected datasets without direct environmental interaction, has emerged as a prominent research direction (Liu et al., 2024b; Chemingui et al., 2025).

Typical offline safe RL approaches predominantly rely on static datasets collected under predefined safety constraints, employing methods such as Lagrangian constraint optimization (Le et al., 2019) and Hamilton-Jacobi (HJ) reachability analysis (Zheng et al., 2024) to derive policies with robust safety guarantees. Despite their effectiveness, these methods exhibit limited adaptability in realistic

---

\*Corresponding Author

multi-task scenarios involving multiple safety constraints (i.e., multiple cost functions) or dynamically changing safety budgets. For instance, in autonomous driving tasks, vehicles must respect diverse constraints, including lane-dependent speed limits (multi-constraint) and varying fuel consumption budgets along a driving trajectory (multi-budget) (Kiran et al., 2021). Addressing these multi-task settings via independently trained individual policies is computationally prohibitive and inefficient, as it neglects the intrinsic structural similarities across tasks. This exacerbates learning complexity, particularly in practical offline safe RL contexts, where available data is inherently scarce (Dulac-Arnold et al., 2021; Gu et al., 2022).

Leveraging knowledge-sharing mechanisms, multi-task learning approaches (Zhang & Yang, 2021; Gronauer & Diepold, 2022) have notably enhanced sample efficiency across interrelated tasks. Transformer architectures (Vaswani et al., 2017; Islam et al., 2024), celebrated for their scalability and in-context learning ability, have demonstrated remarkable versatility in fields such as natural language processing (Kalyan et al., 2021), computer vision (Khan et al., 2022), and decision-making (Yang et al., 2023). Recent developments have also expanded Transformers into offline safe RL, incorporating explicit constraint representations such as Cost-To-Go (CTG) tokens (Zhang et al., 2023; Liu et al., 2023) or logic tokens (Guo et al., 2024). These tokens analogously extend the Return-To-Go (RTG) formalism introduced in DT (Chen et al., 2021; Li et al., 2023), facilitating autoregressive policy training under explicitly modeled safety constraints. Despite significant progress, these approaches currently face limitations in adaptive safety within complex multi-task environments: they inadequately distinguish between diverse constraint specifications, resulting in reduced adaptability when cost functions differ across tasks. Additionally, inherent tensions between maximizing rewards and maintaining safety force these methods to rely extensively on predefined RTG objectives derived from prior knowledge, thus restricting dynamic adjustment capabilities in response to evolving safety budgets during policy execution, and ultimately constraining their flexibility.

To overcome these limitations, we introduce **Co**nstraint **P**rioritized **D**ecision **T**ransformer (CoPDT), a novel DT-based framework designed for offline safe RL that exhibits enhanced adaptability to multiple safety constraints and dynamically changing budgets in multi-task settings. Specifically, we propose a constraint prioritized prompt encoder, which explicitly partitions trajectory data into safe and unsafe segments guided by observed cost signals, individually encoding each segment to effectively capture constraint-specific characteristics. This approach enables CoPDT to clearly differentiate among diverse constraints, facilitating adaptive decision-making across varying conditions. Moreover, we introduce a constraint prioritized Return-To-Go (CPRTG) token mechanism, dynamically generating RTG targets based on the constraint-specific encodings and current CTG tokens. Unlike traditional methods reliant upon static RTG inputs, the proposed CPRTG mechanism achieves enhanced flexibility by adaptively accommodating changing safety budgets, thereby producing more robust and effective policies. Extensive experiments on the OSRL benchmark (Liu et al., 2024b) demonstrate that CoPDT significantly surpasses existing DT-based approaches in multi-task safety adaptability, successfully satisfying constraints in over twice as many tasks, and exhibiting superior efficacy in safe transfer learning scenarios.

## 2 Preliminaries

### 2.1 Multi-task offline safe RL

Safe RL is typically modeled as a Constrained Markov Decision Process (CMDP), defined as a tuple $\langle S, A, r, c, P, \gamma, b \rangle$. Here, $S$ and $A$ denote the state and action spaces, respectively; $r : S \times A \to [-R_{\max}, R_{\max}]$ and $c : S \times A \to \{0, 1\}$ represent the reward and binary cost functions. $P : S \times A \times S \to [0, 1]$ specifies the transition dynamics, $\gamma \in (0, 1)$ is the discount factor, and $b$ is the safety threshold. A policy $\pi : S \to \Delta(A)$ maps states to action distributions. Under policy $\pi$, the expected discounted reward return and cost return are defined as $R(\pi) = \mathbb{E}_{\tau \sim P_\pi} \left[ \sum_{t=0}^{\infty} \gamma^t r(s_t, a_t) \right]$ and $C(\pi) = \mathbb{E}_{\tau \sim P_\pi} \left[ \sum_{t=0}^{\infty} \gamma^t c(s_t, a_t) \right]$, where $\tau = (s_0, a_0, s_1, a_1, \dots) \sim P_\pi$ denotes a trajectory induced by $\pi$ and the environment dynamics $P$. Thus, the objective of solving a CMDP is to find a policy that maximizes reward return while ensuring the cost return remains below the safety threshold:

$$\max_{\pi} R(\pi), \quad s.t. \ C(\pi) \leq b. \tag{1}$$

In multi-task offline safe RL, a policy $\pi$ must effectively adapt across multiple safety budgets or constraints. Each task is formulated as a distinct CMDP, characterized by variations in elements

such as states $S$, actions $A$, transition dynamics $P$, reward functions $r$, cost functions $c$, and safety thresholds (budgets) $b$. Specifically, **multi-budget** scenarios involve tasks that differ primarily in their safety threshold $b$, where, for example, a lower budget ($b = 10$) enforces more conservative behavior compared to a higher budget ($b = 100$), which allows more aggressive policies. In contrast, **multi-constraint** scenarios entail variations in the cost functions themselves; for instance, one task may employ a cost function $c_1 = \mathbb{I}_{\text{condition}}(v > 1.0)$, whereas another task adopts $c_2 = \mathbb{I}_{\text{condition}}(v > 2.0)$, where $\mathbb{I}_{\text{condition}}(\cdot)$ is the indicator function and $v$ denotes a state variable such as velocity. Additionally, other CMDP components $(S, A, P, r)$ may also vary across tasks. During training, the policy is exposed to a collection of tasks $\{\mathcal{T}_j\}_{j=1}^M$ and their offline datasets $\{\mathcal{D}_j\}_{j=1}^M$, while at deployment, it must generalize effectively to a given task $\mathcal{T}$. If $\mathcal{T} \in \{\mathcal{T}_j\}_{j=1}^M$, the policy receives a **single** expert trajectory for task identification in multi-constraint settings. Otherwise, for efficient transfer to an unseen task, it is provided with $L$ expert trajectories from task $\mathcal{T}$.

## 2.2 Decision Transformer for offline safe RL

Decision Transformer (DT) is one of the most prominent methods that apply sequence modeling to decision-making. It uses a Transformer framework, modeling RL's reward maximization problem as a sequence prediction task. When applied to offline safe RL, DT models the trajectory as the following to support training and generation with Transformers:

$$\tau = (\hat{C}_1, \hat{R}_1, s_1, a_1, \hat{C}_2, \hat{R}_2, s_2, a_2, \ldots, \hat{C}_T, \hat{R}_T, s_T, a_T), \tag{2}$$

where $\hat{R}_t = \sum_{i=t}^T r_i$ is the Return-To-Go (RTG) token at time step $t$, and $\hat{C}_t = \sum_{i=t}^T c_i$ is the Cost-To-Go (CTG) token. Let $\tau_{-K:t} = (\hat{C}_{t-K}, \hat{R}_{t-K}, s_{t-K}, a_{t-K}, \ldots, \hat{C}_{t-1}, \hat{R}_{t-1}, s_{t-1}, a_{t-1})$, DT's policy can be expressed as $\pi_{\text{DT}}(\hat{a}_t | \tau_{-K:t}, \hat{C}_t, \hat{R}_t, s_t)$, inferring the current action based on the previous K-step trajectory, the current RTG, CTG and state. The policy is trained by minimizing the difference between the inferred actions $\hat{a}_t$ and actions $a_t$ in offline datasets. During deployment, the DT's policy $\pi_{\text{DT}}$ requires an initial RTG token $\hat{R}_1$, CTG token $\hat{C}_1$ (which is exactly the safety threshold $b$), and state $s_1$ to generate actions, with the RTG and CTG updated using $\hat{R}_{t+1} = \hat{R}_t - r_t$ and $\hat{C}_{t+1} = \hat{C}_t - c_t$.

# 3 Method

This section gives the detailed CoPDT, a novel algorithm for adaptable multi-task offline safe RL (Figure 1). Section 3.1 presents CoPDT's procedure for prompt encoding, Section 3.2 illustrates the process of CPRTG token generation, while Section 3.3 introduces CoPDT's overall algorithm.

## 3.1 Constraint prioritized prompt encoder learning

To enable CoPDT to effectively adapt to tasks with varying constraints, the policy must accurately identify these constraints. Therefore, we introduce a prompt encoder that extracts relevant information from task trajectories to generate task representations, helping distinguish between constraints. **Environment-specific Encoders** First, considering the presence of tasks with different state action spaces, using a single unified neural network for all tasks becomes challenging due to the inconsistency in input dimensions. Therefore, we categorize different tasks into distinct environments, where tasks within the same environment share identical $(S, A, P)$. Then, for each environment, we apply environment-specific encoders to align the dimensionality of states and actions across tasks. Specifically, for the $i$-th environment $\mathcal{E}_i$, we introduce two encoders, $e_{s,i}$ for states and $e_{a,i}$ for actions, along with decoders $d_{s,i}$ and $d_{a,i}$. To ensure that the action encodings retain sufficient information from the original actions, $e_{a,i}$ and $d_{a,i}$ are trained using the reconstruction loss:

$$\min_{e_{a,i}, d_{a,i}} \mathbb{E}_{a_t \sim \mathcal{D}_i}[(d_{a,i}(e_{a,i}(a_t)) - a_t)^2], \tag{3}$$

where $\mathcal{D}_i$ represents the combined offline dataset for all tasks within environment $\mathcal{E}_i$ and $a_t$ is the sampled action. As for $e_{s,i}$ and $d_{s,i}$, we introduce an additional inverse dynamics model $g_i$, and train them by simultaneously minimizing the reconstruction error and the inverse dynamics error to incorporate both state information and dynamics transition information into state encodings:

$$\min_{e_{s,i}, d_{s,i}, g_i} \mathbb{E}_{s_t, a_t, s'_t \sim \mathcal{D}_i}[(d_{s,i}(e_{s,i}(s_t)) - s_t)^2 + (g_i(e_{s,i}(s_t), e_{s,i}(s'_t)) - e_{a,i}(a_t))^2], \tag{4}$$

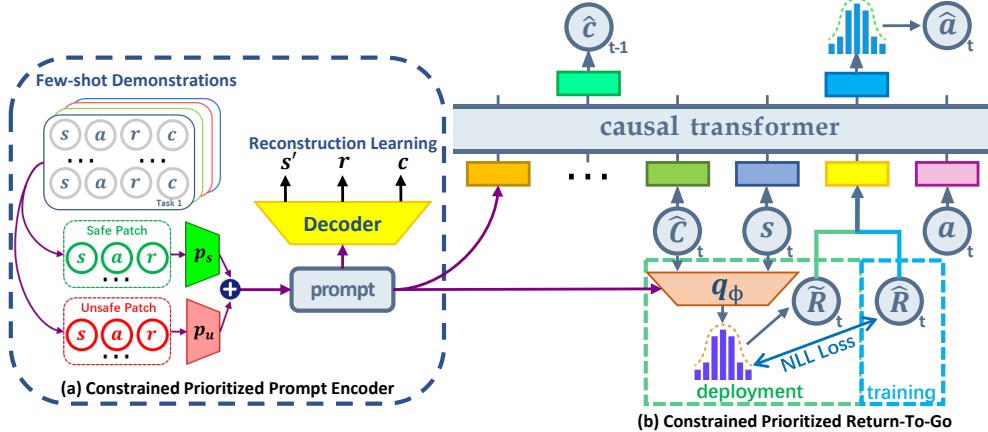

Figure 1: Structure of CoPDT.

where $s_t, a_t, s'_t$ are the sampled state-action transitions.

In previous DT-based methods for handling same-environment tasks with varying rewards, the typical method Prompt-DT (Xu et al., 2022c) uses limited $K$-step trajectory segments as prompts for task representation. However, due to the sparse nature of costs, short trajectory segments may not contain sufficient information for constraint recognition. Therefore, new methods are needed to more effectively extract cost-related information from the trajectories.

**Constraint Prioritized Prompt Encoder** To address the challenge posed by the sparse nature of $c$, we propose the constraint prioritized prompt encoder $p_e$. Specifically, $p_e = (p_s, p_u)$ consists of two sub-prompt encoders. Given a reference trajectory $\tau^* = (s_1, a_1, r_1, c_1, \ldots, s_T, a_T, r_T, c_T)$ for task $\mathcal{T}$, where $r_t, c_t$ are the reward and cost of time step $t$, the prompt encoding $z$ is computed as follows:

$$z = p_e(\tau^*) = \frac{1}{T} \sum_{t=1}^{T} (\mathbb{I}_{\text{condition}}(c_t = 0) p_s(s_t, a_t, r_t) + \mathbb{I}_{\text{condition}}(c_t = 1) p_u(s_t, a_t, r_t)), \quad (5)$$

where $\mathbb{I}_{\text{condition}}$ is the indicator function. Since task differences may arise from variations in state spaces (environments), reward functions, and cost functions, it is crucial for $p_e$ to capture information from all three factors to ensure accurate task differentiation. To accomplish this, we introduce three additional decoder networks $f_s$, $f_r$, and $f_c$, and train them by minimizing prediction errors:

$$\min_{p_e, f_s, f_r, f_c} \mathbb{E}_{\mathcal{T} \sim \{\mathcal{T}_j\}_{j=1}^{M}} [\mathbb{E}_{\tau^*, s_t, a_t, s'_t, r_t, c_t \sim \mathcal{D}_\mathcal{T}} [(f_s(s_t, a_t, p_e(\tau^*)) - s'_t)^2$$
$$+ (f_r(s_t, a_t, p_e(\tau^*)) - r_t)^2 + (f_c(s_t, a_t, p_e(\tau^*)) - c_t)^2]], \quad (6)$$

where $\mathcal{D}_\mathcal{T}$ is the dataset for task $\mathcal{T}$, containing trajectories with both reward and cost information, $\tau^*$ refers to the trajectory that includes $s'_t, r_t, c_t$.

The constraint prioritized prompt encoder exploits the binary nature of cost signals to decouple cost information from the prompt encoder's input and uses it to select the appropriate sub-encoder network. This design enables efficient utilization of cost information by capturing task distinctions through variations in the input distributions of states, actions, and rewards across different encoders. The resulting prompt encoding $z$ serves as part of the input of the DT policy for task identification.

### 3.2 Constraint prioritized RTG token generation

Leveraging constraint-level representations from the prompt encoder and budget-level signals from the CTG, a unified DT policy can potentially support adaptable multi-task offline safe learning. However, prior methods often overlook a key issue in multi-budget settings: the conflict between reward and cost objectives, which are represented by the RTG token $\hat{R}_t$ and the CTG token $\hat{C}_t$ in DT, respectively. Handling varying budgets requires adapting to diverse CTG inputs, but selecting a suitable RTG under changing CTGs remains challenging. When RTG values are misaligned, the agent tends to prioritize rewards due to its limited ability to resolve the conflict between RTG and

CTG, risking safety violations. Therefore, our goal is to automatically generate a conflict-free RTG based on the identified constraint representation and the current CTG.

**Modeling RTG Conditioned on CTG**    To achieve the generation of conflict-free RTGs in CoPDT for different budgets, a straightforward approach is to model RTG as conditioned on CTG, i.e., learning the model $p(\hat{R}_t|\hat{C}_t)$ from the offline data. Since the relationship between RTG and CTG is primarily derived from offline trajectories, where multiple RTGs might correspond to the same CTG, we further constrain the generation process by incorporating state information at each time step, and model $p$ as a non-deterministic normal distribution $\mathcal{N}$ approximated by a neural network $q_\phi$. Formally, given the offline data $\mathcal{D} = \{(s_t, a_t, s'_t, r_t, c_t, \hat{R}_t, \hat{C}_t, t)_k\}_{k=1}^{|\mathcal{D}|}$, we have:

$$q_\phi(\cdot|\hat{C}_t, s_t) = \mathcal{N}(\mu_\phi(\hat{C}_t, s_t), \Sigma_\phi(\hat{C}_t, s_t)), \tag{7}$$

where $\mu_\phi$ and $\Sigma_\phi$ are the mean and standard deviation networks, respectively, and $\hat{C}_t$ and $s_t$ represent the CTG and state at step $t$. To maximize the probability of generating $\hat{R}_t$ conditioned on the given $\hat{C}_t$ and $s_t$, the model $q_\phi$ is optimized by the following negative log-likelihood objective:

$$\min_{q_\phi} \mathbb{E}_{s_t, \hat{R}_t, \hat{C}_t \sim \mathcal{D}}[-\log q_\phi(\hat{R}_t|\hat{C}_t, s_t)]. \tag{8}$$

**CTG-based $\beta$-quantile Sampling for Safe And Expert Inference**    However, such modeling only prioritizes CTG without considering the need for expert-level inferences after ensuring safety. Therefore, in addition to maximizing $p(\hat{R}_t|\hat{C}_t, s_t)$, we also aim to maximize $p(\hat{R}_t|G_t, \hat{C}_t, s_t)$ by introducing a variable $G_t$ that indicates the trajectory is expert after time step $t$. Similar to MGDT (Lee et al., 2022), we apply Bayes' theorem to obtain the following:

$$p(\hat{R}_t|G_t, \hat{C}_t, s_t) \propto p(\hat{R}_t|\hat{C}_t, s_t)p(G_t|\hat{R}_t, \hat{C}_t, s_t), \tag{9}$$

where $p(G_t|\hat{R}_t, \hat{C}_t, s_t)$ represents the probability that the future trajectory is expert given the current RTG, CTG, and state. Intuitively, when fixing $\hat{C}_t$, a higher probability is attributed to $p(G_t|\hat{R}_t, \hat{C}_t, s_t)$ if $\hat{R}_t$ possesses a larger value. Therefore, this term could be maximized by sampling large $\hat{R}_t$. However, when $\hat{R}_t$ becomes excessively large, the conditional probability $p(\hat{R}_t|\hat{C}_t, s_t)$ tends to decrease. To optimize the joint likelihood $p(\hat{R}_t|G_t, \hat{C}_t, s_t)$, it is therefore essential to select an appropriate $\hat{R}_t$ that strikes a suitable balance. To this end, we adopt a quantile-based selection strategy. Specifically, we first sample $X$ values from $q_\phi(\cdot|\hat{C}_t, s_t)$, and then choose the $\beta$-quantile among them as the final RTG token. To find a balanced $\hat{R}$, we propose the CTG-based $\beta$ decay:

$$\beta_t = \min(\beta_{\text{start}} + (\beta_{\text{start}} - \beta_{\text{end}})\frac{\hat{C}_t - \hat{C}_1}{\hat{C}_1}, \beta_{\text{end}}), \tag{10}$$

where $\hat{C}_1$ is the initially given safety threshold, $\beta_{\text{start}}$ and $\beta_{\text{end}}$ are two hyperparameters. When CTG is large—indicating more room for potential future safety violations—a larger $\beta_t$ for more aggressive decision-making is acceptable. Conversely, when CTG is small, the policy should be more conservative, resulting in a smaller $\beta_t$.

**Overall Generation Process**    In conclusion, at time step $t$, we sample $X$ candidate values from $q_\phi(\cdot|\hat{C}_t, s_t)$, and chose the $\beta_t$-quantile value as the CPRTG token, denoted as $\tilde{R}_t$. This token provides a simple but efficient method for adjusting policy conservatism while attaining high-rewarding behaviors during deployment. If the policy does not meet safety requirements, lowering $\beta_{\text{start}}$ or $\beta_{\text{end}}$ can increase conservatism without altering model parameters. Similarly, adjustments can be made to improve reward return when the policy is too conservative. In practice, we typically fix $\beta_{\text{start}}$ as 0.99 and adjust $\beta_{\text{end}}$ only. Additionally, the past trajectory $\tau_{-K:t}$ is incorporated into $q_\phi$ to improve inference accuracy and maintain greater consistency with training. Additional explanations and theoretical analyses of CPRTG from the perspective of offline RL are provided in Appendix A.

### 3.3  Overall algorithm

With the design above, we can apply CoPDT to multi-task scenarios to learn safe policies. Below, we briefly outline CoPDT's training and deployment. Detailed pseudo-codes are provided in Appendix B and the approach for task identification in unknown environments is provided in Appendix C.

During training, CoPDT first learns the environment-specific encoders by Equation (3) and Equation (4) to align the dimensionalities of state, action spaces. Then, the constrained prioritized prompt encoder $p_e$ is trained via Equation (5) to produce a prompt encoding $z$ that captures constraint-relevant information. Given the prompt encoding $z$ as an additional input, the CPRTG generator $q_\phi$ is then similarly optimized by Equation (8) to generate suitable RTGs under various constraints and budgets. With all components prepared, we finally learn the DT policy. Let the policy network be denoted as $\pi_\theta$, which consists of two output heads: $\pi_{\theta,a}$ for actions and $\pi_{\theta,c}$ for costs. The additional cost head is designed to help the policy capture cost-related patterns, thereby improving task identification. Given the expert trajectory $\tau^*$, the learned $p_e$, environment ID $i$ (for choosing environment-specific encoders), and a sampled trajectory $\tau_{-K:t}, \hat{R}_t, s_t$ from task $\mathcal{T}$'s offline dataset, the input can be represented as $o_t = (\tau_{-K:t}, \hat{C}_t, \hat{R}_t, s_t, p_e(\tau^*), i)$. The cost output head $\pi_{\theta,c}$ is modeled deterministically, while the action output head is modeled as a normal distribution:

$$\pi_{\theta,a}(\cdot|o_t) = \mathcal{N}(\mu_{\theta,a}(o_t), \Sigma_{\theta,a}(o_t)), \tag{11}$$

where $\mu_{\theta,a}$ and $\Sigma_{\theta,a}$ are the mean and standard deviation networks for the action output head, respectively. We optimize the policy by minimizing the negative log-likelihood loss and negative entropy loss of the actions, as well as the difference between the predicted costs and true costs:

$$\min_{\pi_{\theta,a}, \pi_{\theta,c}} \mathbb{E}_{\mathcal{T},i\sim\{\mathcal{T}_j\}_{j=1}^M}[\mathbb{E}_{\tau^*,\tau_{-K:t},\hat{C}_t,\hat{R}_t,s_t,a_t,c_t\sim\mathcal{D}_\mathcal{T}}[-\log\pi_{\theta,a}(a_t|o_t)$$
$$-\lambda_h H[\pi_{\theta,a}(\cdot|o_t)] + \lambda_c(\pi_{\theta,c}(o_t) - c_t)^2]], \tag{12}$$

where $H$ is the Shannon entropy regularizer (Haarnoja et al., 2018), $\lambda_h$ and $\lambda_c$ are two hyperparameters that control the weighting of the entropy regularization and the cost loss.

**Deployment**  During deployment, the initial task safety threshold $\hat{C}_1$ is provided, and in each time step, the CPRTG $\tilde{R}_t$ is computed to replace the original RTG $\hat{R}_t$. At this point, the policy's input is $o_t = (\tilde{\tau}_{-K:t}, \hat{C}_t, \tilde{R}_t, s_t, p_e(\tau^*), i)$, where

$$\tilde{\tau}_{-K:t} = (\hat{C}_{t-k}, \tilde{R}_{t-k}, s_{t-k}, a_{t-k}, \ldots, \hat{C}_{t-1}, \tilde{R}_{t-1}, s_{t-1}, a_{t-1}). \tag{13}$$

## 4    Experiments

In this section, we present our experimental analysis conducted on 26 tasks from the OSRL (Liu et al., 2024b) dataset to answer the following questions: (1) Can CoPDT outperform other baselines across various multi-task settings (Section 4.2)? (2) How the design of CoPDT contributes to its performance (Section 4.3)? (3) Can multi-task learning bring about benefits, and whether each component of CoPDT contribute effectively to the performance (Section 4.4)? For page limits, additional experimental results will be provided in Appendix F.

### 4.1    Baselines and tasks

To evaluate the performance of CoPDT, we conduct experiments across several baselines in various multi-task settings. We first compare CoPDT to **CPQ** (Xu et al., 2022a), a widely used single-task offline safe RL method based on conservative estimation, and to **CDT** (Liu et al., 2023), the SOTA DT-based method designed for multi-budget decision-making. Additionally, we consider **FISOR** (Zheng et al., 2024) and **LSPC** (Koirala et al., 2025), two recent baselines that integrate hard constraint modeling and demonstrates SOTA safety performance in single-task settings. To assess performance in more general multi-constraint scenarios, we extend CDT in two ways: (1) by training it on multiple constraints, resulting in **MTCDT**, and (2) by incorporating expert trajectory segments as prompts (Xu et al., 2022c), referred to as **Prompt-CDT**. These baselines are evaluated under three settings: **Single-constraint Multi-budget**, **Multi-constraint Single-budget**, and **Multi-constraint Multi-budget**, where each method aims to handle four distinct budget levels $[10, 20, 40, 80]$ per constraint in multi-budget settings. For training, single-task baselines like CPQ are trained separately for each budget level, while FISOR, which does not support budget conditioning, is trained four times independently and evaluated under the same protocol to ensure fair comparison. In multi-constraint settings, a unified policy is trained for each method to handle all 26 OSRL tasks simultaneously.

The tasks selected from the OSRL dataset used in our experiments consist of 16 navigation tasks and 10 velocity tasks. Specifically, the navigation tasks involve two types of robots (Point and Car)

Table 1: Overall normalized rewards and costs. Each value is averaged over 20 evaluation episodes, 3 random seeds, and the given budgets in each setting. **Bold**: Safe agents. Gray: Unsafe agents. **Blue**: Safe agent with the highest reward in each setting.

| Task | Single-constraint Multi-budget | | | | | | | | | | Multi-constraint Single-budget | | | | | | Multi-constraint Multi-budget | | | | | |
|---|---|---|---|---|---|---|---|---|---|---|---|---|---|---|---|---|---|---|---|---|---|---|
| | CPQ | | CDT | | FISOR | | LSPC | | CoPDT | | MTCDT | | Prompt-CDT | | CoPDT | | MTCDT | | Prompt-CDT | | CoPDT | |
| | r↑ | c↓ | r↑ | c↓ | r↑ | c↓ | r↑ | c↓ | r↑ | c↓ | r↑ | c↓ | r↑ | c↓ | r↑ | c↓ | r↑ | c↓ | r↑ | c↓ | r↑ | c↓ |
| PointButton1 | 0.67 | 5.28 | 0.54 | 5.16 | 0.08 | 1.30 | 0.16 | 1.90 | **0.05** | **0.66** | 0.48 | 4.66 | 0.57 | 6.02 | **0.02** | **0.44** | 0.49 | 4.17 | 0.55 | 4.90 | **0.04** | **0.55** |
| PointButton2 | 0.53 | 6.04 | 0.45 | 4.32 | 0.11 | 1.41 | 0.17 | 1.70 | 0.14 | 1.41 | 0.43 | 4.65 | 0.43 | 4.32 | 0.09 | 1.43 | 0.38 | 3.81 | 0.40 | 4.22 | **0.08** | **0.98** |
| PointCircle1 | **0.41** | **0.94** | **0.55** | **0.55** | 0.44 | 5.54 | 0.55 | 6.64 | **0.50** | **0.63** | **0.51** | **0.52** | **0.53** | **0.88** | 0.55 | 1.19 | **0.52** | **0.47** | **0.55** | **0.87** | 0.55 | 1.09 |
| PointCircle2 | 0.23 | 5.40 | 0.61 | 1.33 | 0.71 | 6.21 | 0.62 | 5.31 | **0.61** | **0.98** | 0.61 | 2.84 | 0.58 | 2.96 | 0.58 | 2.14 | 0.61 | 3.13 | 0.58 | 2.68 | 0.57 | 1.75 |
| PointGoal1 | **0.58** | **0.48** | 0.67 | 1.71 | 0.66 | 2.14 | **0.25** | **0.27** | 0.36 | 0.56 | 0.60 | 1.20 | 0.71 | 1.66 | **0.16** | **0.27** | 0.61 | 1.28 | 0.68 | 1.68 | **0.24** | **0.30** |
| PointGoal2 | 0.39 | 3.45 | 0.54 | 2.84 | 0.29 | 1.28 | **0.25** | **0.85** | 0.31 | 1.02 | 0.46 | 2.30 | 0.55 | 3.34 | **0.19** | **0.60** | 0.45 | 2.01 | 0.54 | 2.94 | **0.26** | **0.66** |
| PointPush1 | 0.23 | 1.60 | 0.27 | 1.42 | **0.31** | **0.89** | 0.13 | 0.97 | 0.19 | 0.88 | 0.24 | 1.14 | 0.24 | 1.49 | **0.08** | **0.50** | 0.23 | 1.11 | 0.24 | 1.25 | **0.12** | **0.69** |
| PointPush2 | 0.16 | 1.42 | 0.20 | 1.76 | 0.24 | 1.40 | **0.11** | **0.89** | 0.19 | 1.47 | 0.22 | 1.93 | 0.18 | 1.53 | 0.11 | 1.28 | 0.20 | 1.77 | 0.17 | 1.49 | **0.11** | **0.83** |
| CarButton1 | 0.48 | 15.40 | 0.20 | 3.97 | **-0.06** | **0.16** | -0.07 | 1.37 | **0.07** | **0.74** | 0.21 | 5.42 | 0.29 | 7.03 | 0.03 | 1.25 | 0.23 | 4.61 | 0.29 | 6.38 | **0.04** | **0.89** |
| CarButton2 | 0.29 | 19.32 | 0.14 | 4.70 | **-0.02** | **0.40** | -0.15 | 1.52 | -0.02 | 1.33 | 0.24 | 5.58 | 0.25 | 6.01 | -0.01 | 1.28 | 0.22 | 5.19 | 0.25 | 5.46 | **-0.02** | **0.94** |
| CarCircle1 | -0.04 | 4.69 | 0.55 | 4.03 | 0.69 | 5.35 | 0.45 | 3.50 | 0.51 | 3.34 | 0.50 | 3.11 | 0.49 | 3.49 | 0.42 | 2.51 | 0.55 | 3.47 | 0.51 | 3.39 | 0.50 | 2.89 |
| CarCircle2 | 0.45 | 1.31 | 0.63 | 6.28 | 0.51 | 4.13 | **0.26** | **0.00** | **0.28** | **0.98** | 0.56 | 7.44 | 0.55 | 6.11 | 0.29 | 1.81 | 0.56 | 6.37 | 0.57 | 5.61 | 0.34 | 1.67 |
| CarGoal1 | 0.76 | 2.29 | 0.64 | 2.13 | **0.43** | **0.72** | 0.24 | 0.51 | **0.33** | **0.47** | 0.54 | 1.39 | 0.57 | 1.90 | **0.17** | **0.26** | 0.54 | 1.48 | 0.56 | 1.80 | **0.22** | **0.32** |
| CarGoal2 | 0.57 | 4.72 | 0.42 | 2.59 | **0.07** | **0.27** | 0.12 | 0.59 | **0.19** | **0.81** | 0.32 | 1.90 | 0.40 | 3.05 | 0.12 | 1.20 | 0.30 | 1.93 | 0.38 | 2.70 | **0.13** | **0.91** |
| CarPush1 | 0.03 | 1.07 | **0.29** | **0.98** | 0.25 | 0.43 | 0.22 | 0.86 | 0.20 | 0.67 | 0.23 | 1.16 | 0.25 | 1.33 | **0.14** | **0.47** | **0.25** | **0.84** | 0.25 | 0.93 | 0.18 | 0.48 |
| CarPush2 | 0.16 | 7.50 | 0.18 | 2.30 | **0.13** | **0.59** | 0.09 | 0.93 | 0.07 | 0.73 | 0.16 | 2.16 | 0.14 | 2.52 | **0.04** | **0.57** | 0.18 | 2.31 | 0.17 | 2.27 | **0.06** | **0.62** |
| SwimmerVelocityV0 | **0.09** | **0.99** | 0.71 | 1.32 | -0.04 | 0.31 | 0.71 | 2.58 | 0.63 | 1.29 | 0.71 | 6.54 | **0.71** | **0.75** | 0.69 | 0.99 | 0.72 | 7.48 | **0.72** | **0.75** | 0.69 | 0.84 |
| SwimmerVelocityV1 | 0.15 | 1.40 | 0.65 | 1.21 | -0.04 | 0.14 | **0.52** | **0.77** | 0.44 | 0.87 | 0.61 | 0.38 | **0.66** | **0.68** | 0.60 | 0.76 | 0.62 | 0.48 | **0.66** | **0.68** | 0.61 | 0.74 |
| HopperVelocityV0 | 0.04 | 2.01 | **0.84** | **0.92** | 0.30 | 0.23 | 0.97 | 8.19 | 0.84 | 1.50 | 0.63 | 15.08 | 0.83 | 5.11 | 0.63 | 6.45 | 0.68 | 13.37 | 0.89 | 5.01 | 0.57 | 4.28 |
| HopperVelocityV1 | 0.15 | 1.49 | 0.72 | 1.60 | **0.16** | **0.86** | 0.97 | 1.24 | 0.35 | 1.17 | 0.67 | 1.31 | 0.70 | 6.85 | 0.16 | 1.07 | 0.68 | 1.13 | 0.68 | 5.50 | 0.27 | 1.09 |
| HalfCheetahVelocityV0 | 0.40 | 2.24 | 0.94 | 1.05 | **0.89** | **0.00** | **0.96** | **0.46** | 0.51 | 0.36 | 0.88 | 15.50 | 1.08 | 38.91 | **0.63** | **0.60** | 0.89 | 14.71 | 1.08 | 35.24 | **0.70** | **0.36** |
| HalfCheetahVelocityV1 | 0.38 | 2.20 | **0.98** | **0.93** | 0.89 | 0.00 | 0.97 | 1.24 | **0.84** | **1.00** | 0.94 | 1.02 | **0.95** | **0.30** | 0.78 | 1.39 | 0.94 | 1.16 | **0.95** | **0.41** | 0.75 | 1.22 |
| Walker2dVelocityV0 | **0.04** | **0.46** | 0.29 | 1.91 | 0.11 | 1.11 | 0.40 | 4.39 | 0.32 | 2.90 | 1.27 | 26.63 | 0.86 | 14.59 | 0.34 | 4.88 | 1.25 | 24.51 | 0.81 | 14.30 | 0.35 | 4.44 |
| Walker2dVelocityV1 | 0.03 | 0.36 | **0.79** | **0.09** | 0.53 | 0.80 | 0.78 | 0.05 | 0.73 | 0.42 | **0.78** | **0.28** | 0.77 | 0.19 | 0.71 | 0.99 | **0.79** | **0.26** | 0.76 | 0.13 | 0.66 | 0.73 |
| AntVelocityV0 | -0.94 | 0.00 | 0.90 | 0.95 | 0.77 | 0.00 | **0.95** | **0.60** | 0.90 | 0.84 | 0.93 | 1.41 | 0.96 | 2.93 | 0.96 | 5.40 | 0.93 | 1.40 | 0.96 | 2.56 | 0.95 | 4.89 |
| AntVelocityV1 | -1.01 | 0.00 | 0.97 | 0.81 | 0.89 | 0.00 | **0.98** | **0.33** | 0.98 | 1.75 | **0.99** | **0.94** | **0.99** | **0.85** | 0.92 | 3.53 | **0.99** | **0.88** | **0.99** | **0.71** | 0.92 | 3.42 |
| Average | 0.19 | 3.54 | 0.56 | 2.19 | 0.36 | 1.37 | 0.42 | 1.88 | **0.40** | **1.11** | 0.57 | 4.48 | 0.59 | 4.81 | **0.36** | **1.66** | 0.57 | 4.20 | 0.58 | 4.38 | **0.38** | **1.45** |

across four distinct scenarios: Button, Circle, Goal, and Push, with two tasks per scenario. On the other hand, the velocity tasks cover five different robots—Ant, HalfCheetah, Hopper, Swimmer, and Walker2d—with two tasks for each robot. Additional details can be found in Appendix D.

## 4.2 Competitive results

In this section, we present the overall performance of CoPDT and all baselines across the three evaluation settings, and the results are summarized in Table 1. First, in the Single-constraint Multi-budget setting, CPQ exhibits poor performance, primarily due to its limited capacity to handle multi-budget datasets, which often include unsafe trajectories. In contrast, FISOR demonstrates robust safety performance, highlighting the effectiveness of its hard constraint modeling. However, both CPQ and FISOR fail to support adaptation to multiple budgets via a unified policy. On the other hand, the Transformer-based method CDT, which leverages the DT architecture, successfully extends the policy to handle multiple budgets simultaneously, achieving competitive reward performance. Nevertheless, its reliance on manually specified RTGs hinders its adaptability, leading to suboptimal safety outcomes due to conflicts between RTGs and CTGs. In comparison, our method, CoPDT, demonstrates better safety performance than LSPC and comparable results to FISOR, while being able to seamlessly adapt to multiple budgets using a unified policy, effectively resolving conflicts between RTGs and CTGs.

Next, in the multi-constraint settings, MTCDT experiences further degradation in safety performance, primarily due to its inability to distinguish between different constraints. In contrast, Prompt-CDT, which incorporates additional expert trajectory segments for constraint identification, achieves marginally better results, making safe decisions in a few more tasks compared to MTCDT. However, the limited availability of cost signals in the expert trajectory segments restricts their informativeness, leading to insufficient constraint-awareness and ultimately suboptimal performance. In contrast, our method CoPDT delivers consistent and significant improvements in safety performance across both single-budget and multi-budget settings, demonstrating the effectiveness of the constraint prioritized prompt encoder in accurately identifying and adapting to a variety of task-specific constraints. Detailed standard deviations are provided in Appendix F.15.

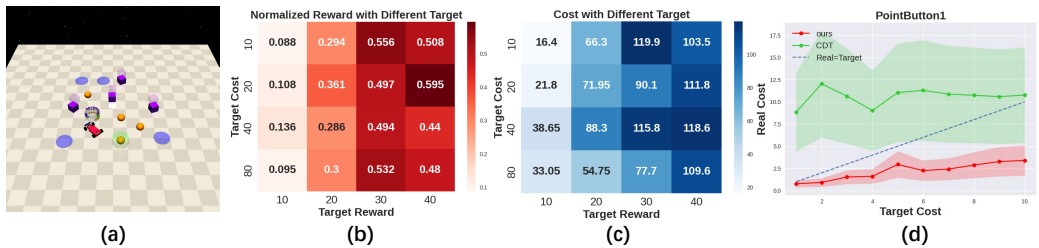

Figure 2: (a) Visualization of task PointButton. (b) The normalized rewards of CDT with different initial RTGs and CTGs. (c) The real costs of CDT with different initial RTGs and CTGs. (d) The safety performance of CoPDT and CDT across various budgets.

## 4.3 Visualization experiment

To reveal how our method CoPDT works, we here design visualization experiments to show the inherent conflict between reward and cost, and assess whether the proposed CPRTG mechanism can generate conflict-free RTGs that enable safe and adaptive decision-making under varying budget constraints. Specifically, we first investigate the conflict between CTGs and RTGs objectives in DT by analyzing CDT's behavior under four different target rewards and four budgets in PointButton1 (Figure 2 (a)). As illustrated in Figure 2 (b) and (c), we observe that both the reward and cost of the DT policy vary significantly with changes in the target reward. In contrast, the cost remains relatively insensitive to changes in the target cost. This asymmetric behavior indicates a conflict between reward and cost objectives, where the DT policy tends to prioritize reward, often at the expense of safety. These findings indicate the necessity of generating suitable RTGs to achieve safe decision-making under different budget constraints. To further quantitatively reveal how the prioritization of CTGs influences safety, we compare the performance of CoPDT and CDT under ten different budgets. As shown in Figure 2 (d), CoPDT consistently satisfies safety constraints across all budgets, with cumulative cost decreasing in accordance with stricter budgets. In contrast, CDT exhibits severe safety violations, especially under stricter budget constraints, illustrating that CoPDT's CPRTG can effectively generate reasonable RTGs based on the CTG, ensuring the higher priority of CTG and enabling safe decision-making under multi-budget scenarios. More results can be seen in Appendix F.2-Appendix F.6.

We design additional experiments to further assess our constraint prioritized prompt encoder's ability in constraint identification. First, we visualize the projection of the generated prompt encodings, as shown in Figure 3 (a), the encodings corresponding to different tasks are well-separated, indicating strong task discriminability. Next, to verify that the constraint prioritized prompt en-

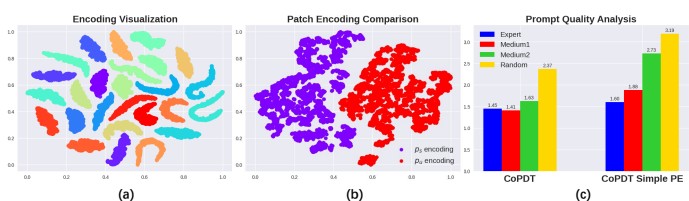

Figure 3: (a) Visualization of prompt encodings. (b) Comparison of safe patch encodings and unsafe patch encodings. (c) Prompt quality analysis on costs.

coder can effectively utilize cost signals for task identification, we conduct a visualization experiment in the PointCircle environment. In this experiment, we first randomly generat 5,000 $(s, a, r)$ samples without including the cost $c$. Each of these samples is then encoded using both $p_s$ and $p_u$, followed by dimensionality reduction. The visualization results are shown in Figure 3 (b). As can be observed, the encoded representations from $p_s$ and $p_u$ exhibit clear differences, indicating that the cost label $c$ significantly influences the encoding process. This demonstrates that the constraint prioritized prompt encoder effectively decouples the learned representations from the behavior policy. Finally, we conduct a prompt quality analysis to assess the encoder's robustness under varying prompt conditions, comparing our encoder with a simple MLP encoder across four prompt qualities: Expert, Medium1, Medium2, and Random. Medium1 and Medium2 correspond to prompts consisting of unseen trajectories with high costs and low rewards. As shown in Figure 3 (c), our encoder maintains

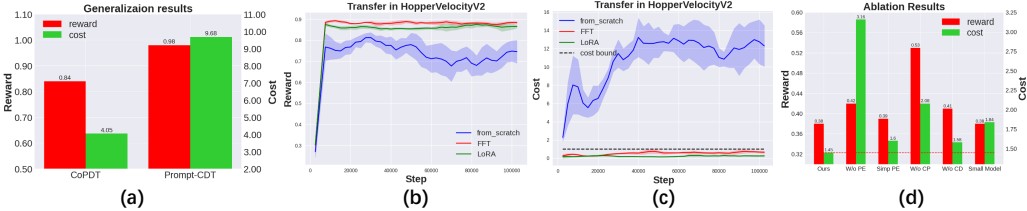

Figure 4: (a) Zero-shot generalization results. (b) Transfer rewards in HopperVelocityV2. (c) Transfer costs in HopperVelocityV2. (d) Main ablation results.

more stable performance across these degraded prompt conditions, demonstrating its robustness even under noisy or suboptimal inputs. Further details can be found in Appendix F.6 and Appendix F.13.

### 4.4 Benefits of multi-task learning and ablations

Since CoPDT is trained in a multi-task setting, we here design experiments to validate its potential for promoting knowledge sharing. Specifically, we first evaluate its zero-shot generalization to unseen constraints (unseen velocity limits) across three robot types. As shown in Figure 4 (a), we can find that CoPDT outperforms Prompt-CDT in terms of average safety generalization, suggesting that its learned task representations capture richer, and our design could cover more transferable information for policy learning. However, a noticeable performance gap remains when compared to policies trained directly on task-specific offline data, indicating the need for few-shot transfer learning. To this end, we introduce 10 expert trajectories under a fixed budget of 10 as additional transfer data and compare two fine-tuning methods—full fine-tuning (FFT) and LoRA (Hu et al., 2022)—against learning from scratch. As shown in Figure 4 (b) and (c), regardless of the fine-tuning method, multi-task pretraining significantly enhances the policy's performance across both reward and cost metrics, highlighting the effectiveness of CoPDT 's multi-task learning in enabling efficient adaptation to novel, yet related tasks. Additional transfer results are presented in Appendix F.7.

Finally, we conduct ablation studies on all selected OSRL tasks in the Multi-constraint Multi-budget setting to evaluate the contributions of key components in CoPDT. The following variants are considered: (1) **W/o PE** omits the constraint prioritized prompt encoder. (2) **Simp PE** replaces the encoder with a simple MLP without separating safe and unsafe patches. (3) **W/o CP** removes the CPRTG module. (4) **W/o CD** disables CTG-based $\beta$ decay. (5) **Small Model** utilizes a DT backbone with fewer parameters. As shown in Figure 4 (d), the extremely poor safety performance of W/o PE highlights the necessity of effective constraint identification, while the higher safety violations in Simp PE further underscore the effectiveness of our constraint prioritized prompt encoder design. The performance degradation observed in W/o CP and W/o CD emphasizes the critical roles of the CPRTG module and the CTG-based $\beta$ decay mechanism in generating reliable RTGs that effectively balance reward and safety. Finally, the drop in safety performance seen in the Small Model reinforces the importance of model capacity, highlighting the necessity of using Transformers to ensure scalability. Additional results can be found in Appendix F.10-Appendix F.14.

## 5 Related work

**Safe RL.** Safe RL ensures policy deployment under safety constraints in addition to reward maximization, which is often formulated as constrained optimization problems (Garcıa & Fernández, 2015; Wachi et al., 2024), and typically solved using Lagrangian multiplier methods (Wachi et al., 2024). These methods learn a cost value function and a parameterized multiplier, adjusting the multiplier based on cumulative cost to enforce safety (Chow et al., 2018; Stooke et al., 2020; Tessler et al., 2019). However, they will involve unsafe real-world interactions during training, making them impractical. To address this, recent work focuses on offline safe RL, learning safe policies from pre-collected data to avoid unsafe exploration. These methods evaluate safety conservatively by treating out-of-distribution (OOD) samples as unsafe (Le et al., 2019; Xu et al., 2022a; Zheng et al., 2024; Yao et al., 2024), thereby reducing extrapolation errors (Fujimoto et al., 2019).

**Transformers in RL.** Transformers have shown impressive performance in complex sequential tasks like large language models (Zhao et al., 2023), inspiring their use in offline RL. DT (Chen et al., 2021), for example, employs a GPT-like Transformer (Achiam et al., 2023) with historical sequences and RTG tokens to predict optimal actions, breaking traditional RL paradigms and circumventing extrapolation errors directly. Extensions such as MGDT (Lee et al., 2022) and Prompt-DT (Xu et al., 2022c) adapt DT to multi-task settings using visual inputs or expert trajectory prompts. For safe RL, CDT (Liu et al., 2023) introduces CTG tokens, while SDT (Guo et al., 2024) uses signal temporal logic tokens to encode richer safety information. However, these methods overlook the differing priorities between RTG and safety tokens, leaving the trade-off between reward and safety constraints unresolved, which motivates our work. More related work is discussed in Appendix G.

## 6 Final remarks

In this work, we propose CoPDT, a novel algorithm for learning adaptable, multi-task offline safe policies within the DT architecture. CoPDT first introduces a constraint prioritized prompt encoder to utilize sparse binary cost information for accurate task identification under different constraints. Then, the CPRTG mechanism is designed to generate reasonable RTGs across varying budgets, effectively resolving conflicts between CTGs and RTGs. Extensive experiments across diverse multi-task settings demonstrate the superior adaptability and safety performance of CoPDT. Looking forward, scaling policy capacity and training across broader task distributions and larger datasets present a promising path toward enhancing knowledge transfer. Furthermore, achieving zero-cost exploration in safe RL using large language models (Wang et al., 2024) and deploying safe RL in embodied robots (Liu et al., 2024a; Feng et al., 2025) are also highly promising future research directions.

## Acknowledgments

This work is supported by the AI & AI for Science Project of Nanjing University, the National Science Foundation of China (62495093,62506159, U24A20324), the Natural Science Foundation of Jiangsu (BK20241199, BK20243039). We thank the anonymous reviewers for their support and helpful discussions on improving the paper.

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

# A  Additional interpretations of CPRTG from the perspective of offline RL

A fundamental challenge in offline RL lies in mitigating extrapolation errors that arise when the policy encounters out-of-distribution (OOD) regions (Prudencio et al., 2023). While DT partially address this issue by constraining policy learning to the support of the offline dataset through supervised learning, they remain vulnerable to OOD generalization when RTG values are treated as also part of the input state. In particular, RTG values that deviate from those observed in the offline data can still induce distributional shift, leading to unsafe or suboptimal behavior. The CPRTG mechanism addresses this by generating RTG targets that remain within the empirical support of the offline dataset, thereby reducing extrapolation risk and enhancing safety. From this perspective, we can make some theoretical analysis about the advantage of using CPRTG.

**Lemma A.1.** *(Janner et al., 2019) Suppose we have two distributions $p_1(x,y) = p_1(x)p_1(y|x)$ and $p_2(x,y) = p_2(x)p_2(y|x)$. We can bound the total variation distance (TVD) of the joint as*

$$D_{TV}(p_1(x,y)||p_2(x,y)) \leq D_{TV}(p_1(x)||p_2(x)) + \mathbb{E}_{x \sim p_1}[D_{TV}(p_1(y|x)||p_2(y|x))]. \tag{14}$$

**Lemma A.2.** *(Janner et al., 2019) Suppose the expected TVD between two dynamics distributions is bounded as $\max_t \mathbb{E}_{s \sim p_1^t(s)}[D_{TV}(p_1(s'|s,a)||p_2(s'|s,a))] \leq \epsilon_m$, and $\max_S D_{TV}(\pi_1(a|s)||\pi_2(a|s)) \leq \epsilon_\pi$, where $p_1^t(s)$ is the state distribution of $\pi_1$ under dynamics $p_1(s'|s,a)$. Then the returns are bounded as:*

$$|\eta_1 - \eta_2| \leq \frac{2R_{max}\gamma(\epsilon_\pi + \epsilon_m)}{(1-\gamma)^2} + \frac{2R_{max}\epsilon_\pi}{1-\gamma}, \tag{15}$$

*where $\eta_i$ is the expected reward return under $\pi_i$ and $p_i$, $\gamma$ is the shared discount factor and $R_{max}$ is the maximum possible reward.*

The two Lemmas mentioned above, although both derived from MBPO (Janner et al., 2019), do not require any additional assumptions for the policy training process (such as being model-based or online training). They only measure the performance of the policy during deployment. Therefore, we can shift our perspective and treat the CTG and RTG distributions from offline data as the true distributions given by the environment. In this context, the policy during deployment can be viewed as being rolled out in a learned model of that environment. This environment model learns state transitions accurately, but the CTG and RTG transitions are not perfectly accurate. According to Lemma A.2, we can conclude that as the CTG and RTG transitions become more aligned with their true distributions, the return obtained by the policy in the learned model will more closely resemble the return in the real environment, which means, in our case, the return during policy deployment will be more similar to the return observed in the offline data. (Under the following theorem and proof, we ignore the past trajectory $\tau_{-K:t}$ for simplification. After incorporating $\tau_{-K:t}$, it can similarly be regarded as part of the state. As a result, the proof remains unchanged—this modification simply introduces $\tau_{-K:t}$ as an additional condition in all probability terms.)

**Theorem A.3.** *Suppose the transition distribution of CTG given the next state during deployment is $p_1(\hat{C}_{t+1}|s',s,\hat{R}_t,\hat{C}_t,a)$, and that induced from the offline dataset is $p_2(\hat{C}_{t+1}|s',s,\hat{R}_t,\hat{C}_t,a)$. The transition distribution of RTG given the next state and next CTG during deployment is $p_1(\hat{R}_{t+1}|\hat{C}_{t+1},s',s,\hat{R}_t,\hat{C}_t,a)$, and that induced from the offline dataset is $p_2(\hat{R}_{t+1}|\hat{C}_{t+1},s',s,\hat{R}_t,\hat{C}_t,a)$. Let the TVD between the CTG transition distribution $D_{TV}(p_1(\hat{C}_{t+1}|s',s,\hat{R}_t,\hat{C}_t,a)||p_2(\hat{C}_{t+1}|s',s,\hat{R}_t,\hat{C}_t,a))$ be $TV(C,t)$, and the TVD between the RTG transition distribution $D_{TV}(p_1(\hat{R}_{t+1}|\hat{C}_{t+1},s',s,\hat{R}_t,\hat{C}_t,a)||p_2(\hat{R}_{t+1}|\hat{C}_{t+1},s',s,\hat{R}_t,\hat{C}_t,a))$ be $TV(R,t)$. If*

$$\max_t \mathbb{E}_{s \sim p_1^t(s),s' \sim p_1(\cdot|s,\hat{R}_t,\hat{C}_t,a)}[TV(C,t)] \leq \epsilon_C, \tag{16}$$

$$\max_t \mathbb{E}_{s \sim p_1^t(s),s' \sim p_1(\cdot|s,\hat{R}_t,\hat{C}_t,a),\hat{C}_{t+1} \sim p_1(\cdot|s',s,\hat{R}_t,\hat{C}_t,a)}[TV(R,t)] \leq \epsilon_R, \tag{17}$$

*then we have*

$$\eta_1^R \geq \eta_2^R - \frac{2R_{max}\gamma(\epsilon_\pi + \epsilon_C + \epsilon_R)}{(1-\gamma)^2} - \frac{2R_{max}\epsilon_\pi}{1-\gamma}, \tag{18}$$

$$\eta_1^C \leq \eta_2^C + \frac{2\gamma(\epsilon_\pi + \epsilon_C + \epsilon_R)}{(1-\gamma)^2} + \frac{2\epsilon_\pi}{1-\gamma}, \tag{19}$$

where $p_1^t(s)$ is the state distribution of the learned DT policy in timestep t, $p_1(\cdot|s, \hat{R}_t, \hat{C}_t, a)$ is the dynamics transition distribution of the target task, $\eta_1^R, \eta_1^C$ are the expected reward return and cost return for the learned DT policy during deployment, and $\eta_2^R, \eta_2^C$ is the expected reward return and cost return for the behavior policy under the state, CTG, RTG transition induced from the dataset.

*Proof.* We view RTG $\hat{R}_t$ and CTG $\hat{C}_t$ from a different perspective, rather than the condition, but part of the state. Then, we take the state, RTG and CTG transition distribution induced from the offline dataset $p_2(s', \hat{R}_{t+1}, \hat{C}_{t+1}|s, \hat{R}_t, \hat{C}_t, a)$ as the ground truth transition distribution, but the state, RTG and CTG transition distribution during deployment as the environment model transition.

First, applying Bayes rule we have

$$p_i(s', \hat{R}_{t+1}, \hat{C}_{t+1}|s, \hat{R}_t, \hat{C}_t, a)$$
$$= p_i(s'|s, \hat{R}_t, \hat{C}_t, a)p_i(\hat{C}_{t+1}|s', s, \hat{R}_t, \hat{C}_t, a)p_i(\hat{R}_{t+1}|\hat{C}_{t+1}, s', s, \hat{R}_t, \hat{C}_t, a), \tag{20}$$

$i = 1, 2$, and $p_1(s'|s, \hat{R}_t, \hat{C}_t, a) = p_2(s'|s, \hat{R}_t, \hat{C}_t, a)$ due to the same state transition distribution. Therefore, apply Lemma A.1 we can obtain

$$D_{\text{TV}}(p_1(s', \hat{R}_{t+1}, \hat{C}_{t+1}|s, \hat{R}_t, \hat{C}_t, a)||p_2(s', \hat{R}_{t+1}, \hat{C}_{t+1}|s, \hat{R}_t, \hat{C}_t, a))$$
$$\leq D_{\text{TV}}(p_1(s'|s, \hat{R}_t, \hat{C}_t, a_t)||p_2(s'|s, \hat{R}_t, \hat{C}_t, a))$$
$$+ \mathbb{E}_{s'\sim p_1(\cdot|s, \hat{R}_t, \hat{C}_t, a)}[D_{\text{TV}}(p_1(\hat{R}_{t+1}, \hat{C}_{t+1}|s', s, \hat{R}_t, \hat{C}_t, a)||p_2(\hat{R}_{t+1}, \hat{C}_{t+1}|s', s, \hat{R}_t, \hat{C}_t, a))]$$
$$\leq \mathbb{E}_{s'\sim p_1(\cdot|s, \hat{R}_t, \hat{C}_t, a)}[D_{\text{TV}}(p_1(\hat{C}_{t+1}|s', s, \hat{R}_t, \hat{C}_t, a)||p_2(\hat{C}_{t+1}|s', s, \hat{R}_t, \hat{C}_t, a))$$
$$+ \mathbb{E}_{\hat{C}_{t+1}\sim p_1(\cdot|s', s, \hat{R}_t, \hat{C}_t, a)}[D_{\text{TV}}(p_1(\hat{R}_{t+1}|\hat{C}_{t+1}, s', s, \hat{R}_t, \hat{C}_t, a)||p_2(\hat{R}_{t+1}|\hat{C}_{t+1}, s', s, \hat{R}_t, \hat{C}_t, a))]].$$
$$\tag{21}$$

Since

$$\max_t \mathbb{E}_{s\sim p_1^t(s), s'\sim p_1(\cdot|s, \hat{R}_t, \hat{C}_t, a)}[\text{TV}(C, t)] \leq \epsilon_C, \tag{22}$$

$$\max_t \mathbb{E}_{s\sim p_1^t(s), s'\sim p_1(\cdot|s, \hat{R}_t, \hat{C}_t, a), \hat{C}_{t+1}\sim p_1(\cdot|s', s, \hat{R}_t, \hat{C}_t, a)}[\text{TV}(R, t)] \leq \epsilon_R, \tag{23}$$

and thus

$$\max_t \mathbb{E}_{s\sim p_1^t(s)}[D_{\text{TV}}(p_1(s', \hat{R}_{t+1}, \hat{C}_{t+1}|s, \hat{R}_t, \hat{C}_t, a)||p_2(s', \hat{R}_{t+1}, \hat{C}_{t+1}|s, \hat{R}_t, \hat{C}_t, a))]$$
$$\leq \max_t \mathbb{E}_{s\sim p_1^t(s), s'\sim p_1(\cdot|s, \hat{R}_t, \hat{C}_t, a)}[\text{TV}(C, t)]$$
$$+ \max_t \mathbb{E}_{s\sim p_1^t(s), s'\sim p_1(\cdot|s, \hat{R}_t, \hat{C}_t, a), \hat{C}_{t+1}\sim p_1(\cdot|s', s, \hat{R}_t, \hat{C}_t, a)}[\text{TV}(R, t)]$$
$$\leq \epsilon_C + \epsilon_R. \tag{24}$$

Therefore, treat $p_i(s', \hat{R}_{t+1}, \hat{C}_{t+1}|s, \hat{R}_t, \hat{C}_t, a)$ as the state transition $p_i(s'|s, a)$ in Lemma A.2, we further obtain

$$|\eta_1^R - \eta_2^R| \leq \frac{2R_{\max}\gamma(\epsilon_\pi + \epsilon_C + \epsilon_R)}{(1-\gamma)^2} + \frac{2R_{\max}\epsilon_\pi}{1-\gamma}, \tag{25}$$

$$|\eta_1^C - \eta_2^C| \leq \frac{2\gamma(\epsilon_\pi + \epsilon_C + \epsilon_R)}{(1-\gamma)^2} + \frac{2\epsilon_\pi}{1-\gamma}. \tag{26}$$

Finally, we have

$$\eta_1^R \geq \eta_2^R - \frac{2R_{\max}\gamma(\epsilon_\pi + \epsilon_C + \epsilon_R)}{(1-\gamma)^2} - \frac{2R_{\max}\epsilon_\pi}{1-\gamma}, \tag{27}$$

$$\eta_1^C \leq \eta_2^C + \frac{2\gamma(\epsilon_\pi + \epsilon_C + \epsilon_R)}{(1-\gamma)^2} + \frac{2\epsilon_\pi}{1-\gamma}. \tag{28}$$

$\square$

Theorem A.3 provides an upper bound on the performance gap between the DT policy during deployment and the offline data-driven behavior policy. This gap is primarily influenced by three factors: $\epsilon_\pi$, $\epsilon_C$, and $\epsilon_R$. $\epsilon_\pi$ is mainly determined by the degree of optimization of the policy loss

function, which is difficult to alter. As for $\epsilon_C$, we rely on the generalization ability of the Transformer for the CTG to adapt to different safety thresholds, and thus, we do not wish to modify the initial settings or update method of the CTG. Therefore, a natural approach to improving the lower bound of policy performance is to reduce the value of $\epsilon_R$. In this context, the CPRTG generator in CoPDT can be viewed as a neural network approximation of $p_2(\hat{R}_{t+1}|\hat{C}_{t+1}, s', s, \hat{R}_t, \hat{C}_t, a)$, which helps to lower $\epsilon_R$ during deployment.

Future research could also explore addressing OOD CTG values, for instance, by mapping large initial CTGs (those exceeding the maximum in the offline dataset) to the dataset's maximum, thus ensuring safety while further mitigating OOD-related extrapolation errors.

## B    Practical algorithms

In this part, we will offer the detailed algorithms of CoPDT in multi-constraint scenarios. As described in the main paper, the workflow of CoPDT primarily includes four parts: $q_\phi$ training, $p_e$ training, policy training, and policy deployment. For $q_\phi$ and $p_e = (p_s, p_u)$, we both use simple multi-layer perceptron (MLP) networks, and additional prompt embeddings, environment IDs, and previous K-step trajectories will be used as inputs for $q_\phi$:

$$\min_{q_\phi} \mathbb{E}_{\mathcal{T}, i \sim \{\mathcal{T}_j\}_{j=1}^M}[\mathbb{E}_{\tau^*, \tau_{-K:t}, s_t, \hat{R}_t, \hat{C}_t \sim \mathcal{D}_\mathcal{T}}[-\log q_\phi(\hat{R}_t|\hat{C}_t, s_t, p_e(\tau^*), \tau_{-K:t}, i)]], \qquad (29)$$

where $\tau_{-K:t}$ is concatenated into a single vector, and is used to improve the accuracy of distribution fitting and maintain consistency between the inputs of $q_\phi$ and $\pi_\theta$. For computational efficiency, we adopt a lightweight MLP architecture here. However, a Transformer-based model, similar to the one used in the DT policy, can also be employed to potentially enhance performance. Exploring such a design remains an interesting direction for future research.

The use of $p_s$ and $p_u$ in $p_e$ is similar to traditional context-based meta RL (Rakelly et al., 2019; Li et al., 2021; Yuan & Lu, 2022). Assume that the safe patch, classified using cost information, is represented as $\{(s_t, a_t, r_t)\}_{t=1}^{T_s}$. For each sample $(s_t, a_t, r_t)$ within this patch, we first concatenate it into a single vector $x_t$. Then, $x_t$ is passed through the MLP neural network $p_s$ to obtain an output vector $z_t$. Consequently, we obtain $T_s$ output vectors for the safe patch. Similarly, we can obtain $T_u$ output vectors for the unsafe patch. By averaging these $T_s + T_u$ output vectors, we obtain the final prompt embedding $z$. During the training of $p_e$, the prompt embedding $z$ is further input into MLP networks $f_s$, $f_r$, and $f_c$ to attempt to predict the corresponding $s'$, $r$, and $c$ values based on the given $(s, a)$ information ($f_s$, $f_r$, and $f_c$ are decoupled from the DT policy). The prediction is then used to compute a regression loss, which allows gradients to be backpropagated into $p_s$ and $p_u$. The relationship between $p_e$ and $f_s$, $f_r$, $f_c$ is essentially that of the encoder and decoder in a traditional autoencoder (Zhai et al., 2018). They are trained jointly before DT training, but only the frozen encoder (not updated with DT) is required for the DT policy training and deployment phase.

Detailed pseudo-codes for $q_\phi$ training, $p_e$ training, and policy training are provided in Algorithm 1, while the pseudo-codes for policy deployment are provided in Algorithm 2.

## C    Distinguish constraints in unknown environments

In this section, we will introduce the task (constraint) identification method when the environment ID is unknown. Based on the definition in Section 2.1, an environment is determined by its state space, action space, and dynamics transition. Therefore, we need to infer the true environment ID based on this information. First, we filter out potential candidate environments from the previously seen environments based on the state space and action space dimensions of the unknown environment. Then, we sequentially use the environment-specific encoders, environment-specific decoders, and the inverse dynamics model from the candidate environments to test the given trajectory in the unknown environment. Specifically, given the trajectory $(s_t, a_t, s'_t)_{t=1}^T$, and the set $\{e_{s,i}, e_{a,i}, d_{s,i}, d_{a,i}, g_i\}_{i=1}^{N'}$ of all $N'$ candidate environments, the objective is as follows:

$$\min_i \frac{1}{T} \sum_{t=1}^T [(d_{a,i}(e_{a,i}(a_t)) - a_t)^2 + (d_{s,i}(e_{s,i}(s_t)) - s_t)^2 + (g_i(e_{s,i}(s_t), e_{s,i}(s'_t)) - e_{a,i}(a_t))^2], \quad (30)$$

**Algorithm 1** CoPDT Training
***

**Input:** task set $\{\mathcal{T}_j\}_{j=1}^M$, environment set $\{\mathcal{E}_i\}_{i=1}^N$, offline dataset for each task $\{\mathcal{D}_{\mathcal{T}_j}\}_{j=1}^M$, DT trajectory length $K$, hyperparameters $\lambda_h, \lambda_c$.

**Initialize:** constraint prioritized prompt encoder $p_e$, decoders $f_s, f_r, f_c$, state action encoders and decoders for each environment $\{e_{s,i}, e_{a,i}, d_{s,i}, d_{a,i}, g_i\}_{i=1}^N$, CPRTG generator $q_\phi$, DT policy $\pi_\theta$.

**for** step in environment-specific training steps **do**
    **for** each environment $\mathcal{E}_i$ **do**
        Merge each task dataset in this environment to get the environment dataset $\mathcal{D}_i$.
        Sample a batch $\{(s_t, a_t, s_t')\}$.
        Update $e_{a,i}$ and $d_{a,i}$ with Equation (3).
        Update $e_{s,i}, d_{s,i}$ and $g_i$ with Equation (4).
    **end for**
**end for**
**for** step in prompt encoder training steps **do**
    **for** each task $\mathcal{T}$ with its environment ID $i$ **do**
        Sample a batch $\{(\tau^*, s_t, a_t, s_t', r_t, c_t)\}$ from $\mathcal{D}_\mathcal{T}$.
        Encode states sampled with $e_{s,i}$ and actions sampled with $e_{a,i}$.
        Update $p_e, f_s, f_r, f_c$ with Equation (6).
    **end for**
**end for**
**for** step in policy training steps **do**
    **for** each task $\mathcal{T}$ with its environment ID $i$ **do**
        Sample a batch $\{(\tau^*, \tau_{-K:t}, \hat{C}_t, \hat{R}_t, s_t, a_t, c_t)\}$ from $\mathcal{D}_T$.
        Update $\pi_\theta$ with Equation (12).
        Update $q_\phi$ with Equation (29).
    **end for**
**end for**
Return $\{e_{s,i}, e_{a,i}\}_{i=1}^N, p_e, q_\phi, \pi_\theta$.
***

**Algorithm 2** CoPDT Deployment
***

**Input:** initial CTG $\hat{C}_1$, environment ID $i$, constraint prioritized prompt encoder $p_e$, state action encoders $e_{s,i}, e_{a,i}$, CPRTG generator $q_\phi$, DT policy $\pi_\theta$, expert trajectory $\tau^*$, DT trajectory length $K$, hyperparameters $X, \beta_{\text{start}}, \beta_{\text{end}}$.

**Initialize:** input sequence $\tau = []$.

Encode states and actions in $\tau^*$ with $e_{s,i}$ and $e_{a,i}$.

Compute the prompt encoding $z$ according to Equation (5).

**for** t=1,...,T **do**
    Observe current state $s_t$.
    Compute $\beta_t$ according to Equation (10).
    Sample $X$ values in distribution $q_\phi(\cdot|\tau[-K:], \hat{C}_t, s_t, z, i)$ and select the $\beta_t$-quantile of it as $\tilde{R}_t$.

    Sample action $a_t$ from $\pi_{\theta,a}(\cdot|\tau[-K:], \hat{C}_t, \tilde{R}_t, s_t, z, i)$.
    Step action $a_t$ in the task environment to get $r_t, c_t$.
    Compute $\hat{C}_{t+1} = \hat{C}_t - c_t$.
    Append $\{\hat{C}_t, \tilde{R}_t, s_t, a_t\}$ to $\tau$.
**end for**
***

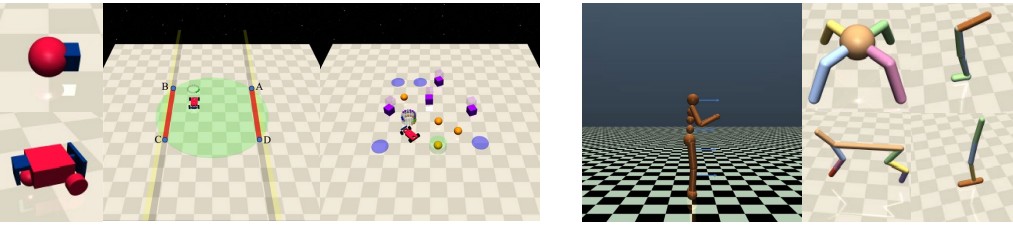

| (a) Navigation Tasks | (b) Velocity Tasks |

Figure 5: Tasks used in this paper. (a) Navigation Tasks based on Point and Car robots. (b) Velocity Tasks based on Ant, HalfCheetah, Hopper, Walker2d, and Swimmer robots.

Table 2: Detailed velocity thresholds for new designed tasks.

| Tasks | Velocity Threshold |
|---|---|
| AntVelocityV2 | 2.52 |
| HalfCheetahVelocityV2 | 3.05 |
| HopperVelocityV2 | 0.56 |
| SwimmerVelocityV2 | 0.18 |
| Walker2dVelocityV2 | 2.00 |

where the first term is the action reconstruction loss, which aims to ensure consistency in the action space; the second term is the state reconstruction loss, which aims to ensure consistency in the state space; and the third term is the inverse dynamics error, which ensures consistency in the dynamics transition. Once the environment ID is determined, we revert to the previous setup, where the trajectory is passed through the environment-specific state encoder, environment-specific action encoder, and the constraint prioritized prompt encoder to obtain the prompt encoding, which serves as the basis for task identification.

# D    Detailed description of the tasks and baselines

## D.1    Tasks and Datasets

All pretraining tasks used in this paper are derived from the Safety-Gymnasium's Navigation Tasks and Velocity Tasks. In the Navigation Tasks, there are two different types of robots: Point and Car, which we need to control to navigate through the environment and earn rewards by reaching target points, pressing the correct buttons, or moving in designated directions. Different tasks also have varying costs, such as avoiding collisions with specific targets, preventing incorrect button presses, and staying within designated boundaries.

The Velocity Tasks are built on traditional MuJoCo (Todorov et al., 2012) simulations, requiring robots such as Ant, HalfCheetah, Swimmer, and Walker2d to move, where higher speeds result in higher rewards. However, each robot has specific safety velocity thresholds for different tasks, and exceeding these thresholds leads to unsafe states. For detailed descriptions of each task, refer to the original Safety-Gymnasium paper (Ji et al., 2023). Besides the mentioned tasks, we designed five new Velocity tasks for task transfer, which differ from previous ones only in their velocity thresholds, as detailed in Table 2.

The offline datasets used for each task are sourced from OSRL (Liu et al., 2024b). Specifically, the datasets for the VelocityV0 and VelocityV2 tasks were additionally collected using OSRL's original data collection methods.

## D.2    Baselines

We provide a more detailed introduction to the baselines of the experiment in this section.

- **CPQ** is a CMDP-based offline safe RL algorithm built upon the classic offline RL method CQL (Kumar et al., 2020). It incorporates the conservative regularization operator from CQL into the cost critic, treating out-of-distribution samples as unsafe. Unlike traditional methods that use Lagrangian multipliers for policy updates, CPQ directly truncates the reward critic to 0 for unsafe state-action pairs, preventing unsafe policy execution. This algorithm has become one of the most common baselines in offline safe algorithms and is considered state-of-the-art in CMDP-based offline safe algorithms.

- **CDT** is the previous SOTA algorithm under sequence modeling for offline safe RL. After incorporating CTG into DT, CDT also seeks to address the conflict between safety constraints and reward maximization. To tackle this issue, CDT proposes a data augmentation approach, where reward returns for certain safe but low-reward trajectories in the offline dataset are re-labeled with higher values. However, the effectiveness of this data augmentation method is still limited by the amount of augmented data and does not fundamentally resolve the underlying issue. In our experiments, all CDT-based methods utilize manually specified initial RTG values for different constraints and budgets, as provided by the original OSRL implementation (authored by the same team behind CDT) [2].

- **FISOR** is the SOTA offline safe RL algorithm based on hard constraint modeling. Similar to other algorithms under hard constraints Yu et al. (2022); Ganai et al. (2023), it first divides the state space into feasible and infeasible regions. It then uses the IQL Kostrikov et al. (2022) algorithm to learn feasible value functions for offline feasible region identification. Next, FISOR sets distinct learning objectives for the feasible and infeasible regions. In the feasible region, it aims to maximize the reward while ensuring feasibility, while in the infeasible region, it focuses on minimizing constraint violations. Finally, FISOR employs a diffusion model to represent and learn the policy. In the experiments, although the policy learning of FISOR is independent of the safety thresholds, operating under a fully hard-constrained setting, all cost results are still normalized using four safety thresholds: $[10, 20, 40, 80]$.

- **LSPC** is a recently proposed hard-constrained offline safe RL algorithm. Similar to FISOR, it employs IQL to learn feasible value functions. Once the feasible value functions are obtained, it minimizes the feasible value as a weighting to train the decoder of a CVAE. The trained CVAE decoder can then be regarded as a policy that minimizes safety violations, referred to as **LSPC-S**. Furthermore, an additional policy $\pi(z|s)$ is trained to generate the latent encoding input $z$ for the CVAE decoder, where $\pi$ is optimized to maximize the expected reward. Finally, a hyperparameter is introduced to control the optimization strength of $\pi$, balancing safety satisfaction and reward maximization. This variant is referred to as **LSPC-O**. In experiments, we adopt LSPC-O, which provides a balanced trade-off between safety and reward, as the baseline, and evaluate it using the same protocol as FISOR.

- **MTCDT** is a straightforward multi-task extension of CDT. It addresses the varying state and action dimensions in cross-environment tasks by utilizing distinct input and output heads for each environment. Additionally, MTCDT attempts to identify different tasks based solely on the sequential inputs of DT for multi-task decision-making.

- **Prompt-CDT** is also a multi-task extension of CDT. Building on MTCDT, Prompt-CDT utilizes additional expert trajectory segments as prompts to assist in task identification.

## E Hyperparameters

The training and deployment of CoPDT both involve the selection of hyperparameters. To ensure reproducibility, this section outlines the specific hyperparameters used in our experiments. CoPDT is implemented based on CDT within the OSRL framework, and the default parameters are retained for any hyperparameters not explicitly mentioned in Table 3.

Table 3: Hyperparameter choices of CoPDT.

| | Hyperparameter | Value |
|---|---|---|
| environment-specific encoders | state encode dim | 32 |
| | action encode dim | 2 |
| | all network hidden layers | [128, 128, 128] |
| | update steps | 100000 |
| | batch size | 2048 |
| | learning rate | 0.0001 |
| prompt encoder $p_e$ | prompt encode dim | 16 |
| | all encoder hidden layers | [256, 256, 256] |
| | all decoder hidden layers | [128, 128] |
| | update steps | 100000 |
| | batch size | 2048 |
| | learning rate | 0.0001 |
| RTG generator $q_\phi$ | environment-specific state input head output dim | 64 |
| | environment-specific state input head hidden layers | [128, 128, 128] |
| | generator hidden layers | [1024, 256, 128] |
| | update steps | 100000 |
| | batch size | 2048 |
| | learning rate | 0.0001 |
| policy learning and deployment | DT embedding dim | 512 |
| | DT num layers | 3 |
| | DT num heads | 8 |
| | DT sequence len | 20 |
| | environment-specific state input head output dim | 64 |
| | environment-specific state input head hidden layers | [128, 128, 128] |
| | environment-specific action input head output dim | 32 |
| | environment-specific action input head hidden layers | [128, 128, 128] |
| | environment-specific action output head input dim | 32 |
| | environment-specific action output head hidden layers | [128, 128, 128] |
| | update steps | 200000 |
| | batch size | 1024 |
| | learning rate | 0.0001 |
| | $\lambda_h$ | 0.1 |
| | $\lambda_c$ | 0.02 |
| | $\beta_{\text{start}}$ | 0.99 |
| | $\beta_{\text{end}}$ | 0.8 |
| | $X$ | 1000 |

Table 4: Time complexity comparison.

| | CoPDT (Single-constraint) | CDT | CoPDT (Multi-constraint) | MTCDT | Prompt-CDT |
|---|---|---|---|---|---|
| Prompt Encoder Training | \ | \ | 1.330 h | \ | \ |
| DT Policy Training | 15.734 h | 15.737 h | 19.584 h | 19.288 h | 33.008 h |
| CPRTG Generator Training | 0.250 h | \ | 1.404 h | \ | \ |
| Deployment | 0.012 s/step | 0.008 s/step | 0.017 s/step | 0.008 s/step | 0.014 s/step |

# F   More experimental results

## F.1   Time complexity analysis

Time complexity during policy training and deployment is a critical issue in real-world applications. Therefore, in this section, we provide the analysis of CoPDT's time complexity. Due to the use of neural networks, it is not feasible to provide a quantitative analysis. Instead, we first compare CoPDT qualitatively with other baseline algorithms and present the specific results through actual data.

---

[2]https://github.com/liuzuxin/OSRL

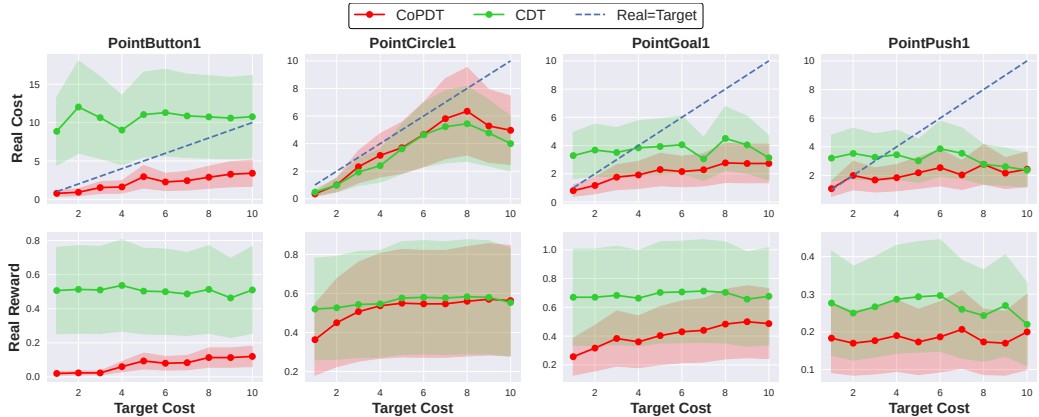

Figure 6: Performance of CoPDT and CDT under various safety thresholds. Target costs and real costs are normalized by 10.

First, during the training process in the Single-constraint setting (no need for constraint identification), the policy training for CoPDT and CDT is identical, with the only difference being in the CPRTG generator's training. Since we use a traditional MLP neural network in the CPRTG, its computational complexity is much lower than that of the large Transformer networks used in policy training, resulting in minimal additional overhead for CoPDT during training. In the Multi-constraint setting, CoPDT, compared to MTCDT and Prompt-CDT, involves not only the CPRTG generator but also the training of the prompt encoder. This training includes both environment-specific encoder training and constraint prioritized prompt encoder training. Similar to CPRTG training, the prompt encoder training only involves MLP networks, so it does not introduce significant additional overhead. The specific data is shown in Table 4. In the Single-constraint scenario, we use the training of a fixed task as the result, and in the Multi-constraint scenario, we report the average training time per task (constraint). During deployment, both Single-constraint and Multi-constraint scenarios use the same task for testing. All experiments were conducted on a single NVIDIA GeForce RTX 4090, and fairness was ensured even in the presence of CPU resource contention. The results in the table align with our earlier analysis, showing that the primary overhead during training comes from policy training, with the additional MLP network training overhead being minimal. In contrast, Prompt-CDT incurs higher training overhead due to the use of sequence-based prompts.

In the policy deployment process, the additional time complexity for CoPDT mainly arises from the use of the CPRTG generator. By observing the last row of Table 4, we can see that in the Single-constraint setting, the use of the CPRTG generator does introduce some extra overhead. In the Multi-constraint setting, the additional overhead increases, as the use of prompts adds computational complexity both during policy inference and in the CPRTG generator, but the additional overhead is still within an acceptable range (smaller than 0.01 second) for real-world deployment. Further analysis of the additional time complexity introduced by the CPRTG generator is provided in Appendix F.10.

### F.2 More results on multi-budget decision making

In this section, we provide more results on the adaptation capability of CoPDT and CDT to different safety thresholds, as shown in Figure 6. It is shown that CoPDT is able to effectively adapt under any safety threshold, ensuring the safety performance of the policy. At the same time, both its cost and reward exhibit a clear increasing trend as the safety threshold rises. Although CDT shows some advantages in terms of reward, it clearly lags behind in terms of safety. In three of the environments, it fails to demonstrate adaptability to different safety thresholds, resulting in poor safety performance when the safety threshold is low. These experimental results validate the strong adaptation capability of CoPDT across different safety thresholds.

### F.3 Deployment phase conservatism adjustment with different $\beta_{\text{end}}$

$\beta_{\text{end}}$ serves as a critical hyperparameter that controls the conservatism of CPRTG generation. To evaluate its impact on policy performance, we conduct a sensitivity analysis, as illustrated in Figure 7. It is evident that as $\beta_{\text{end}}$ increases, both the reward and cost of the policy also gradually rise, indicating that CoPDT can effectively modulate the level of conservatism in decision-making through tuning $\beta_{\text{end}}$, demonstrating its flexibility and adaptability across different safety-reward trade-offs.

Based on the previous analysis, the conservatism of CoPDT can be flexibly adjusted by modifying $\beta_{\text{end}}$. To further explore this adaptability, we introduce a new experimental setting within the Single-Constraint Multi-Budget setting, referred to as Oracle.

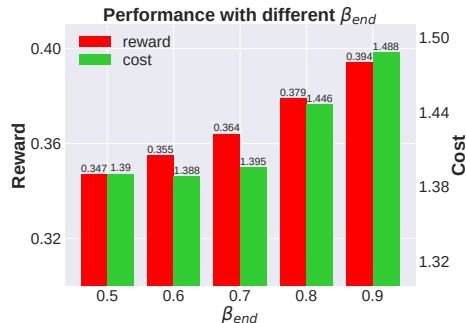

Figure 7: Parameter sensitivity study on $\beta_{\text{end}}$.

In this setting, algorithms are allowed to fine-tune hyperparameters for each task with different constraints and even across varying budgets within the same constraint to achieve optimal performance. The following algorithms are evaluated under this setting: (1) **BC-Safe**: For each constraint and budget, trajectories from the offline dataset that satisfy the corresponding budget are filtered, and behavior cloning is performed based on the filtered safe trajectories. (2) **FISOR**: Separate reverse-expectile $\tau$ values are used for tasks with different constraints. (3) **CoPDT**: Different $\beta_{\text{end}}$ values are applied across tasks with varying constraints.

The experimental results, summarized in Table 5, highlight the strengths and weaknesses of the three algorithms. BC-Safe demonstrates superior average performance but satisfies the safety constraints in the fewest number of tasks. FISOR, on the other hand, ensures compliance with safety constraints across all tasks but shows limited ability to optimize rewards under safety conditions. CoPDT strikes a balance: while it fails to meet safety constraints in a small number of tasks, it exhibits strong reward optimization capabilities under safety constraints, offering greater flexibility. Additionally, a unique advantage of CoPDT over the other two algorithms lies in its ability to adjust conservatism during the deployment phase. Unlike BC-Safe and FISOR, CoPDT allows for adjustments to $\beta_{\text{end}}$ without requiring policy retraining, enabling efficient and flexible tuning.

### F.4 Analysis on stitching ability

In this section, we compare the stitching ability of CoPDT and CDT. Specifically, we construct a challenging test setting by removing all trajectories with cumulative cost $\leq 10$ from the offline datasets of eight Point tasks, and evaluate the policies under a safety budget of 10. As shown in Table 6, CDT fails to learn safe policies in this setting, exhibiting significant safety violations due to the lack of safe trajectories in the offline data. In contrast, CoPDT successfully learns safety-satisfying policies in more than half of the tasks. These results demonstrate that the CPRTG module in CoPDT enables effective trajectory stitching beyond simple behavior cloning, facilitating safe decision even in the absence of directly safe trajectories.

### F.5 Visualization of CPRTGs

In this section, we visualize how $\tilde{R}_t$ changes in response to variations in $\hat{C}_t$ and $\beta_t$, as shown in Figure 8. The results indicate that $\tilde{R}_t$ increases significantly with higher values of $\hat{C}_t$ and also rises notably as $\beta_t$ increases. This confirms the rationale behind modeling RTG under CTG conditions and applying decay to $\beta_t$, allowing the policy to gradually adjust its conservatism based on potential future safety violations.

### F.6 Visualization of prompt encodings

In this section, we additionally explored the encoding process and properties of the constraint prioritized prompt encoder. First, we visualized the state distributions after encoding each environment's state separately using the environment-specific state encoder (Figure 9(a)). The results reveal clear separations between different environments, though tasks within the same environment remain indistinguishable. Next, we visualized the results after applying the prompt encoder (Figure 9(b)). At

Table 5: Overall performance in the Oracle setting.

| Task | Oracle | | | | | |
| --- | --- | --- | --- | --- | --- | --- |
| | BC-Safe | | FISOR | | CoPDT | |
| | r↑ | c↓ | r↑ | c↓ | r↑ | c↓ |
| PointButton1 | **0.04** | **0.74** | -0.01 | 0.28 | 0.09 | 0.91 |
| PointButton2 | 0.15 | 1.75 | **0.05** | **0.43** | 0.08 | 0.92 |
| PointCircle1 | **0.38** | **0.16** | 0.05 | 0.06 | 0.54 | 0.62 |
| PointCircle2 | **0.45** | **0.99** | 0.20 | 0.00 | 0.61 | 0.98 |
| PointGoal1 | **0.38** | **0.53** | 0.03 | 0.01 | 0.51 | 0.87 |
| PointGoal2 | 0.29 | 1.13 | **0.05** | **0.08** | 0.29 | 0.91 |
| PointPush1 | **0.13** | **0.67** | 0.31 | 0.89 | 0.19 | 0.88 |
| PointPush2 | 0.13 | 1.05 | **0.09** | **0.29** | 0.13 | 0.63 |
| CarButton1 | **0.07** | **0.87** | -0.02 | 0.78 | 0.07 | 0.74 |
| CarButton2 | -0.03 | 1.25 | 0.02 | 0.40 | -0.02 | 0.89 |
| CarCircle1 | 0.29 | 1.66 | 0.21 | 0.24 | 0.49 | 2.96 |
| CarCircle2 | 0.51 | 5.17 | 0.40 | 0.42 | 0.28 | 0.98 |
| CarGoal1 | **0.28** | **0.39** | 0.43 | 0.72 | 0.39 | 0.75 |
| CarGoal2 | **0.14** | **0.57** | 0.07 | 0.27 | 0.19 | 0.81 |
| CarPush1 | **0.15** | **0.45** | 0.25 | 0.43 | 0.28 | 0.96 |
| CarPush2 | **0.05** | **0.63** | 0.13 | 0.59 | 0.09 | 0.88 |
| SwimmerVelocityV0 | **0.52** | **0.08** | -0.04 | 0.31 | 0.62 | 0.98 |
| SwimmerVelocityV1 | 0.50 | 0.63 | -0.04 | 0.14 | 0.44 | 0.87 |
| HopperVelocityV0 | 0.50 | 0.25 | 0.30 | 0.23 | 0.18 | 0.52 |
| HopperVelocityV1 | 0.42 | 0.65 | 0.16 | 0.86 | 0.18 | 0.86 |
| HalfCheetahVelocityV0 | 0.92 | 1.11 | 0.89 | 0.00 | 0.67 | 0.38 |
| HalfCheetahVelocityV1 | 0.89 | 0.75 | 0.89 | 0.00 | 0.84 | 1.00 |
| Walker2dVelocityV0 | 0.24 | 1.45 | 0.05 | 0.12 | 0.32 | 2.90 |
| Walker2dVelocityV1 | 0.79 | 0.01 | 0.53 | 0.80 | 0.78 | 0.12 |
| AntVelocityV0 | 0.86 | 0.61 | 0.77 | 0.00 | 0.90 | 0.84 |
| AntVelocityV1 | 0.96 | 0.38 | 0.89 | 0.00 | 0.97 | 1.58 |
| Average | 0.39 | 0.92 | 0.25 | 0.32 | 0.39 | 0.99 |

Table 6: Stitching ability analysis results.

| Task | CDT | | CoPDT | |
| --- | --- | --- | --- | --- |
| | r↑ | c↓ | r↑ | c↓ |
| PointButton1 | 0.56 | 12.66 | **0.01** | **0.88** |
| PointButton2 | 0.50 | 10.28 | 0.04 | 1.49 |
| PointCircle1 | **0.53** | **0.91** | 0.51 | 0.61 |
| PointCircle2 | 0.59 | 1.79 | 0.59 | 1.52 |
| PointGoal1 | 0.74 | 4.65 | **0.25** | **0.98** |
| PointGoal2 | 0.69 | 13.53 | 0.22 | 1.37 |
| PointPush1 | 0.27 | 4.16 | **0.13** | **0.90** |
| PointPush2 | 0.16 | 5.25 | **0.09** | **0.87** |
| Average | 0.51 | 6.65 | 0.23 | 1.08 |

this stage, tasks within the same environment are also successfully differentiated, confirming the effectiveness of our design and loss function selection.

Next, we aim to further explore the properties of the constraint prioritized prompt encoder, specifically whether it focuses more on differences in state-action distribution driven by varying cost information, rather than on the state-action distribution itself. To test this, we specifically selected tasks where the cost function significantly affects the state-action distribution (e.g., PointCircle1 and PointCircle2). Using the policy of PointCircle1, we collected data in both tasks to ensure consistency in state-action distribution for the prompts. The results, as shown in Figure 9(c) and Figure 9(d), reveal that the constraint prioritized prompt encoder effectively distinguishes tasks based on cost information, while

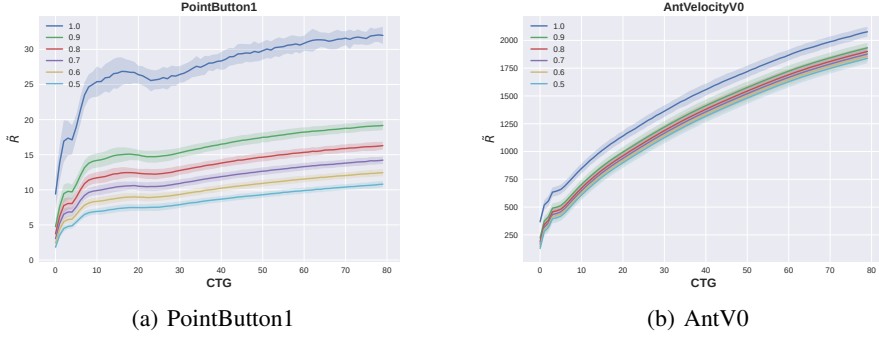

(a) PointButton1          (b) AntV0

Figure 8: Generated $\tilde{R}_t$ under different $\hat{C}_t$ and $\beta_t$.

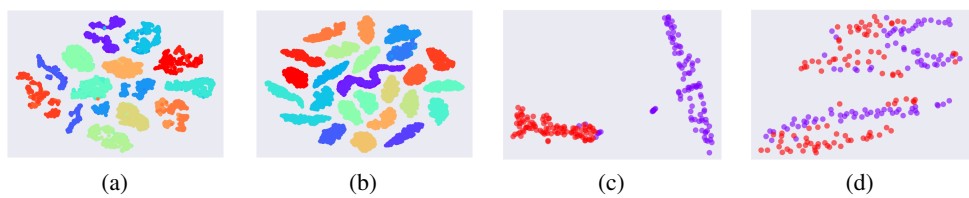

(a)        (b)        (c)        (d)

Figure 9: Different visualization results. (a) Visualization of state encodings after using the environment-specific state encoders. (b) Visualization of prompt encodings of the constraint prioritized prompt encoder using expert trajectories as prompts. (c) Visualization of prompt encodings of the constraint prioritized prompt encoder using trajectories collected by a same behavior policy for PointCircle1 and PointCircle2. (d) Visualization of prompt encodings of the simple MLP encoder using trajectories collected by a same behavior policy for PointCircle1 and PointCircle2.

the traditional MLP encoder suffers from task confusion due to the similar state-action distribution. This highlights the robust cost information extraction ability of the constraint prioritized prompt encoder.

### F.7 More policy transfer results

In this section, we first provide transfer results in more tasks, as shown in Figure 10. In four out of the five tasks, although the CoPDT pretrained model can't guarantee zero-shot safety, fine-tuning with FFT or LoRA shows clear advantages in both safety performance and reward compared to training from scratch. In one environment, the pretrained model ensures zero-shot safety, but fine-tuning results in a performance drop, likely due to insufficient trajectory coverage. Comparing FFT and LoRA, both perform similarly, possibly due to the relatively small model size, with LoRA offering lower fine-tuning overhead. Based on the above results, it can be found that the pretraining of CoPDT can effectively improve the policy's adaptation ability in new tasks, enhancing applicability.

Then, to demonstrate that our prompt encoder design can include more effective information than directly inputting sequence prompts during task transfer, we compare the results with Prompt-CDT under two fine-tuning methods: FFT and LoRA. Additionally, we introduce a new environment, AntCircle, which has a lower similarity to the pretraining task. In AntCircle, the state space, action space, and dynamics transition are consistent with AntVelocity, but both the reward function and cost function undergo significant changes. The reward function is modified to represent the speed at which the Ant robot moves along a circle, while the cost function now penalizes the robot's x-coordinate rather than its speed. The results are shown in Figure 11. First, by observing the results in similar tasks, we see that Prompt-CDT's multitask pretraining also provides a certain level of improvement in task transfer in environments other than HalfCheetah. However, compared to CoPDT, Prompt-CDT still exhibits inferior task transfer performance. In HalfCheetah, it leads to a significantly higher violation of safety constraints. In Hopper, FFT shows a noticeable drop in performance, while LoRA

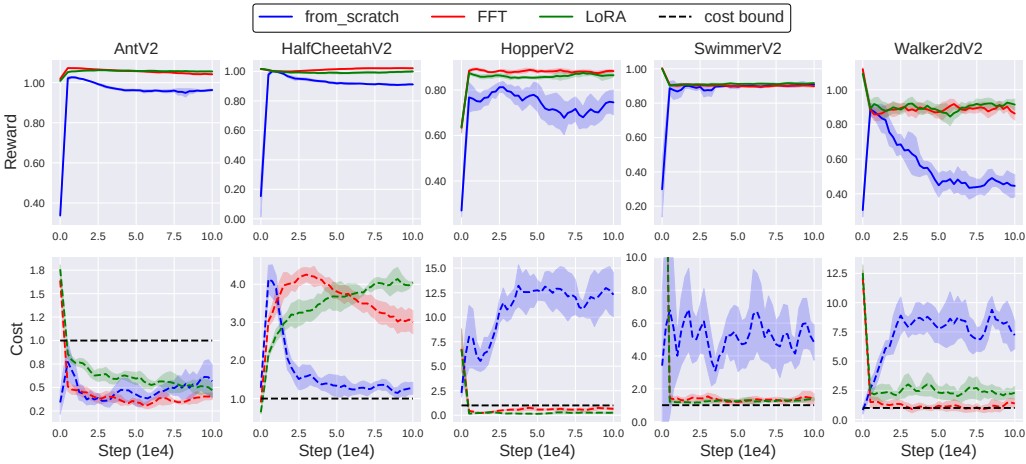

Figure 10: The transfer results of CoPDT in 5 new tasks.

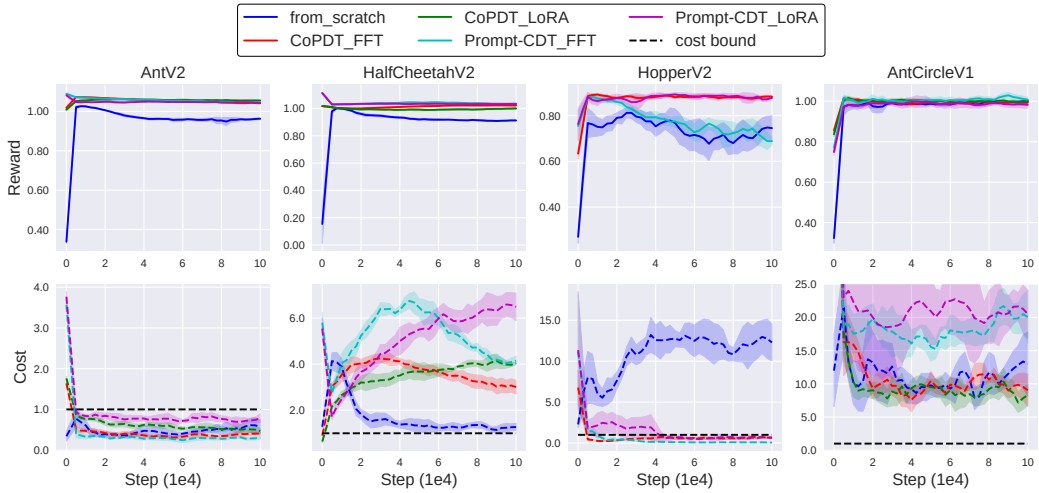

Figure 11: Transfer comparison with Prompt-CDT and transfer results in a dissimilar task.

causes instability in safety. This clearly indicates that CoPDT's use of the prompt encoder provides more effective information for knowledge transfer than directly using sequence prompts.

Next, by observing the results in dissimilar tasks, we find that even in scenarios with low task similarity, CoPDT's pretraining still provides some performance improvement compared to learning from scratch. However, due to the limited amount of transferable knowledge, this improvement is less significant than in similar tasks. In contrast, Prompt-CDT's pretraining results in a noticeable decline in safety performance. This indicates that the sequential prompts used in Prompt-CDT do not always bring additional information gain and may sometimes interfere with the extraction of effective information. Furthermore, these results suggest that in low-similarity scenarios, few-shot adaptation may not always yield stable results, and using larger datasets for training might be a better alternative. Additionally, increasing the diversity of tasks during pretraining is an effective way to enhance the policy's transferability.

Finally, we further evaluated transfer performance on two unseen robots, Ball and Drone, using 10 expert trajectories with a safety budget of 10. Due to the differences in state and action spaces, we first trained new environment-specific encoders for these four tasks, followed by fine-tuning the rest of the CoPDT architecture. As shown in Figure 12, although the parameter-efficient LoRA fine-tuning method performs suboptimally, FFT consistently outperforms training from scratch across all tasks

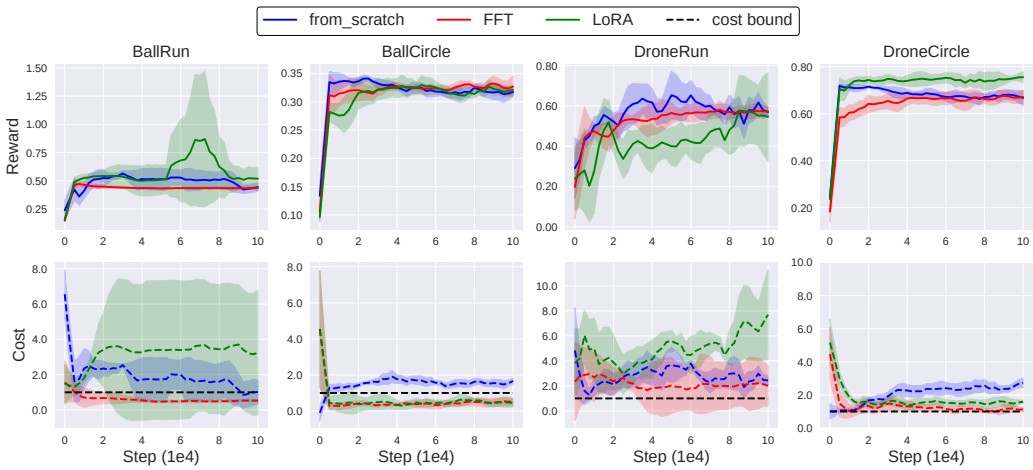

Figure 12: The transfer results of CoPDT to unseen robots.

in terms of safe decision-making. These results further validate that CoPDT's multi-task learning framework enables effective knowledge transfer to unseen tasks under limited data.

## F.8 Discussion on zero-shot constraint generalization

This section gives the detailed generalization results in Figure 4 (a) of the main paper, as shown in Table 7. The results reveal that while CoPDT demonstrates better zero-shot performance compared to Prompt-CDT, it still exhibits substantial safety violations, making it unsuitable for zero-shot deployment. We attribute this primarily to the insufficient number of similar tasks encountered during multi-task training, which fails to support the emergence of robust zero-shot generalization. This finding suggests that in typical decision-making scenarios with limited tasks and data, task transfer remains a more practical and reliable approach than relying on policy generalization for unseen tasks.

Table 7: Zero-shot generalization to tasks with unseen constraints.

| Task | CoPDT | | Prompt-CDT | |
|---|---|---|---|---|
| | reward | cost | reward | cost |
| AntV2 | 0.99 | 2.18 | 1.10 | 5.16 |
| HalfCheetahV2 | **1.02** | **0.01** | 1.15 | 7.74 |
| HopperV2 | 0.52 | 9.96 | 0.70 | 16.14 |

To further support our hypothesis that pretraining on a broader range of similar tasks can enhance the zero-shot generalization capability of the learned policy, we conduct an additional experiment in which policies are trained under a larger number of velocity thresholds (from 2 to 6) in the AntVelocity and HopperVelocity tasks. Specifically, in the 2-threshold setting, the thresholds are [2.57, 2.62] for Ant and [0.37, 0.74] for Hopper. In the 6-threshold setting, we use [2.42, 2.47, 2.55, 2.57, 2.62, 2.67] for Ant and [0.37, 0.42, 0.52, 0.62, 0.67, 0.74] for Hopper. For evaluation, beyond the original zero-shot generalization tasks (AntV2 with threshold 2.52 and HopperV2 with threshold 0.56), we further design two additional zero-shot generalization tasks: AntV7 (threshold 2.4) and HopperV7 (threshold 0.35). Notably, the V2 tasks correspond to interpolation generalization settings, while the V7 tasks represent extrapolation settings. All results are obtained under a budget of 10, and evaluated using 3 random seeds. The experimental results are shown in Table 8. It can be observed that when CoPDT is pre-trained on a larger number of similar tasks—similar to the setup in meta-RL—the model exhibits significant improvements in safety performance on both interpolation (V2) and extrapolation (V7) generalization tasks. The improvement is particularly notable in the Hopper environment, which may be attributed to the large threshold gap between the original two Hopper tasks, making pre-training on only two tasks insufficient. These results support the claim that our constraint prioritized prompt encoder not only captures task-specific memory but also possesses a certain degree of generalization capability to unseen constraints.

Table 8: Zero-shot generalization results with more similar training tasks.

| Task | CoPDT (2 similar tasks) | | CoPDT (6 similar tasks) | |
|---|---|---|---|---|
| | r↑ | c↓ | r↑ | c↓ |
| AntV2 | 0.99 | 2.18 | 0.98 | 1.56 |
| AntV7 | 0.99 | 3.28 | 0.96 | 2.34 |
| HopperV2 | 0.52 | 9.96 | **0.48** | **0.04** |
| HopperV7 | 0.24 | 21.68 | 0.43 | 4.57 |
| Average | 0.69 | 9.28 | 0.71 | 2.13 |

Table 9: Comparison results of coupled and decoupled prompt encoder training.

| Task | Multi-constraint Multi-budget | | | |
|---|---|---|---|---|
| | CoPDT coupled | | CoPDT decoupled | |
| | r↑ | c↓ | r↑ | c↓ |
| PointButton1 | **0.03** | **0.39** | 0.04 | 0.55 |
| PointButton2 | 0.12 | 1.48 | 0.08 | 0.98 |
| PointCircle1 | 0.50 | 0.54 | 0.55 | 1.09 |
| PointCircle2 | 0.51 | 0.58 | 0.57 | 1.75 |
| PointGoal1 | 0.24 | 0.35 | 0.24 | 0.30 |
| PointGoal2 | 0.25 | 1.04 | 0.26 | 0.66 |
| PointPush1 | 0.14 | 0.57 | 0.12 | 0.69 |
| PointPush2 | 0.09 | 0.54 | 0.11 | 0.83 |
| CarButton1 | 0.05 | 0.87 | 0.04 | 0.89 |
| CarButton2 | -0.02 | 0.80 | -0.02 | 0.94 |
| CarCircle1 | 0.50 | 2.92 | 0.50 | 2.89 |
| CarCircle2 | 0.31 | 1.76 | 0.34 | 1.67 |
| CarGoal1 | 0.20 | 0.39 | 0.22 | 0.32 |
| CarGoal2 | 0.15 | 0.98 | 0.13 | 0.91 |
| CarPush1 | 0.17 | 0.31 | 0.18 | 0.48 |
| CarPush2 | 0.06 | 0.85 | 0.06 | 0.62 |
| SwimmerVelocityV0 | 0.68 | 0.82 | 0.69 | 0.84 |
| SwimmerVelocityV1 | 0.59 | 0.59 | 0.61 | 0.74 |
| HopperVelocityV0 | 0.63 | 3.05 | 0.57 | 4.28 |
| HopperVelocityV1 | 0.21 | 1.02 | 0.27 | 1.09 |
| HalfCheetahVelocityV0 | 0.66 | 0.18 | 0.70 | 0.36 |
| HalfCheetahVelocityV1 | 0.69 | 1.73 | 0.75 | 1.22 |
| Walker2dVelocityV0 | 0.39 | 5.25 | 0.35 | 4.44 |
| Walker2dVelocityV1 | 0.73 | 1.15 | 0.66 | 0.73 |
| AntVelocityV0 | 0.96 | 4.60 | 0.95 | 4.89 |
| AntVelocityV1 | 0.86 | 5.10 | 0.92 | 3.42 |
| Average | 0.37 | 1.46 | 0.38 | 1.45 |

## F.9 Discussion on coupled or decoupled prompt encoder training

In context-based multi-task RL, whether the prompt encoder should be trained jointly (coupled) with the policy remains an open research question. Coupled training allows for richer parameter updates and potentially more accurate adaptation, while decoupled training better preserves the desirable properties obtained during pretraining. To investigate this, we conduct experiments comparing the coupled and decoupled training of the prompt encoder within CoPDT. As shown in Table 9, coupling the prompt encoder with policy optimization does not yield significant performance gains. This suggests that, in our setting, the simpler decoupled training scheme is already sufficient. Nevertheless, the broader question of whether and when prompt encoders should be coupled with policy training warrants further investigation.

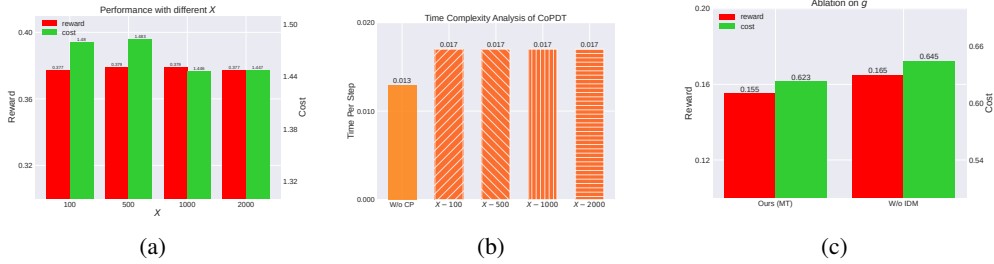

Figure 13: (a) Performance ablation on $X$. (b) Time complexity analysis of $X$. (c) Performance ablation on whether using the inverse dynamics model $g$ in 4 tasks.

### F.10 Ablation on CPRTG sample number $X$

In this section, we first conducted an ablation study on different choices of the CPRTG sample number $X$ to investigate the impact of this hyperparameter on policy performance. The results are shown in Figure 13(a). It can be observed that as $X$ increases from 100 to 2000, there is almost no significant change in the policy's reward, but a noticeable reduction in the policy's cost. This result confirms that as the CPRTG sample number increases, the sampled values are closer to the desired quantile points, leading to better performance and a certain improvement in safety.

Additionally, we performed a further analysis of the time complexity of the CPRTG generation process in CoPDT, with the results presented in Figure 13(b). From the figure, we can draw two conclusions. First, when CPRTG is used, the time consumption does indeed increase compared to not using CPRTG, indicating that the CPRTG generation process introduces additional computational overhead. Second, when $X$ increases from 100 to 2000, the time overhead remains almost unchanged, suggesting that the additional cost brought by CPRTG mainly comes from the inference of the CPRTG generator's neural network, rather than the sampling of quantile points. Therefore, in practice, we can increase $X$ as much as possible to achieve better policy performance without introducing significant additional computational costs.

### F.11 Discussion and ablation on inverse dynamics model $g$

In Section 3.1, we introduced an additional inverse dynamics model $g$ to compute the inverse dynamics error for training the environment-specific state encoders. The primary motivation for using this model is to address tasks with identical state and action spaces but different dynamics transitions. While such tasks have not appeared in our main experiments, they are still quite common (Nagabandi et al., 2019; Eysenbach et al., 2021; Zhang et al., 2024). In these cases, the inverse dynamics error based on the inverse dynamics model can effectively produce different state representations during the environment-specific state encoder learning phase, thereby reducing the learning difficulty for the constraint prioritized prompt encoder. Moreover, as described in Appendix C, when the environment ID is unknown, the inverse dynamics error based on the inverse dynamics model becomes the core method for distinguishing these tasks, making it an essential component. We also conducted an additional ablation study to ensure that the use of the inverse dynamics model does not negatively impact the policy performance, with results shown in Figure 13(c), which aligns with our expectations.

### F.12 Ablation on LoRA rank

In LoRA, performance is mainly influenced by the LoRA rank $r$ and the LoRA $\alpha$ (Hu et al., 2022). Following standard practice, we set $\alpha$ to twice the value of $r$ and conducted experiments on task transfer with various $r$ values. The results, shown in Figure 14, indicate that performance generally improves as $r$ increases, except in the HalfCheetah task, where an anomaly occurred, consistent with previous findings. These results suggest that when the model has relatively few parameters, increasing the number of fine-tuned parameters positively impacts performance.

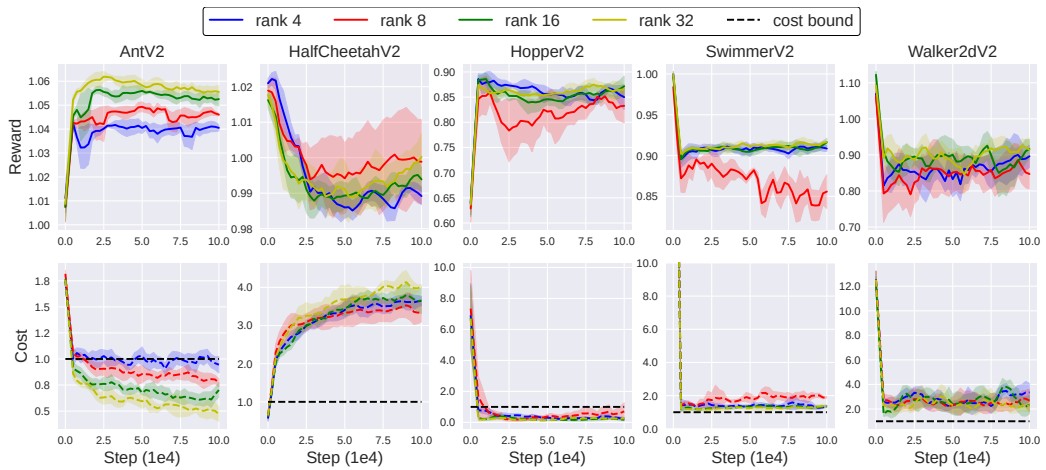

Figure 14: Ablation on different LoRA ranks.

Table 10: Detailed results of the prompt quality analysis done in the Multi-constraint Multi-budget setting. The red results indicate results that are significantly affected by changes in prompt quality.

| Task | CoPDT | | | | | | | | CoPDT Simple PE | | | | | | | | Prompt-CDT | | | | | | | |
|---|---|---|---|---|---|---|---|---|---|---|---|---|---|---|---|---|---|---|---|---|---|---|---|---|
| | Expert | | Medium1 | | Medium2 | | Random | | Expert | | Medium1 | | Medium2 | | Random | | Expert | | Medium1 | | Medium2 | | Random | |
| | r↑ | c↓ | r↑ | c↓ | r↑ | c↓ | r↑ | c↓ | r↑ | c↓ | r↑ | c↓ | r↑ | c↓ | r↑ | c↓ | r↑ | c↓ | r↑ | c↓ | r↑ | c↓ | r↑ | c↓ |
| PointButton1 | 0.04 | 0.55 | 0.05 | 0.48 | 0.05 | 0.69 | 0.08 | 0.68 | 0.02 | 0.60 | 0.03 | 0.53 | 0.04 | 0.50 | 0.03 | 0.48 | 0.55 | 4.90 | 0.56 | 5.03 | 0.56 | 4.64 | 0.55 | 5.31 |
| PointButton2 | 0.08 | 0.98 | 0.04 | 0.93 | 0.08 | 1.05 | 0.07 | 0.87 | 0.10 | 1.21 | 0.00 | 0.68 | 0.02 | 0.69 | 0.00 | 0.64 | 0.40 | 4.22 | 0.41 | 4.21 | 0.38 | 3.79 | 0.46 | 5.06 |
| PointCircle1 | 0.55 | 1.09 | 0.51 | 1.09 | 0.51 | 0.99 | 0.49 | 0.99 | 0.53 | 0.86 | 0.53 | 0.82 | 0.53 | 0.85 | 0.47 | 1.06 | 0.55 | 0.87 | 0.54 | 0.84 | 0.50 | 0.94 | 0.52 | 0.88 |
| PointCircle2 | 0.57 | 1.75 | 0.56 | 1.83 | 0.59 | 2.15 | 0.51 | 2.76 | 0.55 | 1.51 | 0.52 | 1.44 | 0.52 | 1.48 | 0.51 | 2.03 | 0.58 | 2.68 | 0.61 | 3.17 | 0.59 | 2.62 | 0.63 | 3.61 |
| PointGoal1 | 0.24 | 0.30 | 0.25 | 0.34 | 0.24 | 0.31 | 0.24 | 0.37 | 0.26 | 0.36 | 0.25 | 0.40 | 0.26 | 0.32 | 0.23 | 0.34 | 0.68 | 1.68 | 0.67 | 1.47 | 0.66 | 1.58 | 0.68 | 1.76 |
| PointGoal2 | 0.26 | 0.66 | 0.23 | 0.72 | 0.22 | 0.59 | 0.25 | 0.95 | 0.22 | 0.77 | 0.22 | 0.57 | 0.23 | 0.74 | 0.20 | 0.69 | 0.54 | 2.94 | 0.53 | 2.67 | 0.50 | 2.92 | 0.53 | 2.97 |
| PointPush1 | 0.12 | 0.69 | 0.14 | 0.74 | 0.16 | 0.61 | 0.13 | 0.43 | 0.12 | 0.56 | 0.11 | 0.61 | 0.12 | 0.51 | 0.13 | 0.43 | 0.24 | 1.25 | 0.26 | 1.42 | 0.25 | 1.32 | 0.27 | 1.21 |
| PointPush2 | 0.11 | 0.83 | 0.12 | 0.73 | 0.10 | 0.79 | 0.10 | 0.61 | 0.11 | 0.80 | 0.10 | 0.56 | 0.11 | 0.73 | 0.10 | 0.82 | 0.17 | 1.49 | 0.14 | 1.73 | 0.16 | 2.10 | 0.16 | 1.87 |
| CarButton1 | 0.04 | 0.89 | 0.04 | 0.68 | 0.04 | 0.64 | 0.00 | 0.96 | 0.04 | 0.67 | 0.04 | 0.72 | 0.03 | 0.81 | 0.03 | 0.70 | 0.29 | 6.38 | 0.23 | 5.73 | 0.28 | 5.85 | 0.28 | 6.00 |
| CarButton2 | -0.02 | 0.94 | -0.01 | 1.07 | -0.02 | 1.06 | -0.02 | 0.88 | -0.01 | 1.05 | 0.00 | 1.21 | 0.00 | 0.99 | 0.01 | 0.91 | 0.25 | 5.46 | 0.26 | 5.10 | 0.27 | 5.68 | 0.25 | 5.53 |
| CarCircle1 | 0.50 | 2.89 | 0.49 | 2.94 | 0.50 | 2.84 | 0.43 | 2.54 | 0.51 | 3.27 | 0.51 | 3.10 | 0.51 | 3.16 | 0.40 | 2.44 | 0.51 | 3.39 | 0.53 | 3.23 | 0.53 | 3.20 | 0.53 | 3.53 |
| CarCircle2 | 0.34 | 1.67 | 0.33 | 1.75 | 0.32 | 1.70 | 0.50 | 4.92 | 0.32 | 1.37 | 0.31 | 0.95 | 0.31 | 1.12 | 0.41 | 3.19 | 0.57 | 5.61 | 0.58 | 6.65 | 0.57 | 5.92 | 0.56 | 6.14 |
| CarGoal1 | 0.22 | 0.32 | 0.24 | 0.32 | 0.24 | 0.36 | 0.20 | 0.27 | 0.21 | 0.34 | 0.20 | 0.28 | 0.21 | 0.36 | 0.20 | 0.38 | 0.56 | 1.80 | 0.56 | 1.69 | 0.57 | 1.42 | 0.57 | 1.50 |
| CarGoal2 | 0.13 | 0.91 | 0.13 | 0.72 | 0.17 | 1.00 | 0.18 | 1.01 | 0.14 | 0.94 | 0.17 | 0.94 | 0.16 | 0.88 | 0.18 | 0.92 | 0.38 | 2.70 | 0.38 | 2.48 | 0.39 | 2.73 | 0.38 | 2.52 |
| CarPush1 | 0.18 | 0.48 | 0.17 | 0.45 | 0.17 | 0.42 | 0.12 | 0.53 | 0.18 | 0.35 | 0.16 | 0.71 | 0.17 | 0.60 | 0.12 | 0.37 | 0.25 | 0.93 | 0.25 | 0.87 | 0.25 | 0.99 | 0.22 | 1.55 |
| CarPush2 | 0.06 | 0.62 | 0.03 | 0.46 | 0.05 | 0.59 | 0.03 | 0.89 | 0.05 | 0.61 | 0.06 | 0.75 | 0.11 | 2.63 | 0.05 | 2.77 | 0.17 | 2.27 | 0.17 | 2.13 | 0.19 | 2.30 | 0.17 | 2.65 |
| SwimmerVelocityV0 | 0.69 | 0.84 | 0.69 | 0.83 | 0.69 | 0.82 | 0.67 | 10.14 | 0.70 | 1.93 | 0.69 | 1.33 | 0.69 | 1.52 | 0.69 | 2.31 | 0.72 | 0.75 | 0.72 | 0.76 | 0.72 | 0.77 | 0.75 | 15.65 |
| SwimmerVelocityV1 | 0.61 | 0.74 | 0.56 | 0.38 | 0.57 | 0.30 | 0.57 | 0.61 | 0.59 | 0.74 | 0.57 | 0.70 | 0.57 | 0.51 | 0.59 | 0.59 | 0.66 | 0.68 | 0.62 | 0.46 | 0.63 | 0.49 | 0.62 | 0.32 |
| HopperVelocityV0 | 0.57 | 4.28 | 0.51 | 3.96 | 0.43 | 5.63 | 0.19 | 4.75 | 0.55 | 3.25 | 0.57 | 3.32 | 0.55 | 3.36 | 0.38 | 6.89 | 0.89 | 5.01 | 0.93 | 18.86 | 1.00 | 18.04 | 0.90 | 6.29 |
| HopperVelocityV1 | 0.27 | 1.09 | 0.32 | 1.35 | 0.30 | 1.01 | 0.31 | 1.03 | 0.28 | 1.43 | 0.32 | 1.04 | 0.29 | 1.66 | 0.29 | 1.66 | 0.68 | 5.50 | 0.72 | 6.94 | 0.72 | 6.83 | 0.71 | 0.57 |
| HalfCheetahVelocityV0 | 0.70 | 0.36 | 0.68 | 0.54 | 0.69 | 3.98 | 0.75 | 5.73 | 0.70 | 0.38 | 0.82 | 10.29 | 0.99 | 28.97 | 0.98 | 25.89 | 1.08 | 35.24 | 1.06 | 36.06 | 1.05 | 29.64 | 1.09 | 35.71 |
| HalfCheetahVelocityV1 | 0.75 | 1.22 | 0.63 | 0.50 | 0.62 | 0.59 | 0.67 | 0.26 | 0.88 | 0.49 | 0.80 | 0.24 | 0.86 | 0.61 | 0.87 | 0.55 | 0.95 | 0.41 | 0.95 | 0.42 | 0.95 | 0.46 | 0.96 | 0.45 |
| Walker2dVelocityV0 | 0.35 | 4.44 | 0.35 | 4.22 | 0.35 | 4.42 | 0.47 | 9.97 | 0.40 | 5.57 | 0.38 | 5.14 | 0.39 | 5.40 | 0.74 | 13.00 | 0.81 | 14.30 | 0.78 | 13.41 | 1.07 | 18.13 | 1.22 | 23.91 |
| Walker2dVelocityV1 | 0.66 | 0.73 | 0.64 | 0.72 | 0.15 | 1.11 | 0.31 | 1.09 | 0.72 | 0.59 | 0.69 | 0.72 | 0.71 | 0.69 | 0.60 | 1.06 | 0.76 | 0.13 | 0.73 | 0.11 | 0.74 | 0.13 | 0.68 | 0.15 |
| AntVelocityV0 | 0.95 | 4.89 | 0.96 | 5.10 | 0.96 | 5.02 | 0.93 | 4.57 | 0.98 | 8.45 | 0.98 | 8.68 | 0.98 | 8.50 | 0.97 | 8.19 | 0.96 | 2.56 | 0.96 | 2.51 | 0.96 | 2.43 | 0.94 | 1.65 |
| AntVelocityV1 | 0.92 | 3.42 | 0.92 | 3.70 | 0.95 | 3.83 | 0.97 | 3.91 | 0.96 | 3.44 | 0.96 | 3.20 | 0.97 | 3.34 | 1.03 | 5.41 | 0.99 | 0.71 | 0.99 | 0.76 | 0.99 | 0.74 | 1.02 | 1.29 |
| Average | 0.38 | 1.45 | 0.37 | 1.41 | 0.35 | 1.63 | 0.35 | 2.37 | 0.39 | 1.60 | 0.38 | 1.88 | 0.40 | 2.73 | 0.39 | 3.19 | 0.58 | 4.38 | 0.58 | 4.95 | 0.60 | 4.83 | 0.60 | 5.31 |

## F.13 Detailed prompt quality analysis results

In this section, we evaluate the impact of prompt quality on the performance of CoPDT, CoPDT Simple PE, and Prompt-CDT. Specifically, CoPDT Simple PE replaces the constraint prioritized prompt encoder in CoPDT with a standard MLP encoder. We consider four categories of prompt quality: Expert (in-distribution expert trajectories seen during prompt encoder training), Medium1 (unseen trajectories with severe safety violations), Medium2 (unseen trajectories with significantly lower reward performance), and Random (trajectories collected from a random policy). As shown in Table 10, while all methods are affected by prompt quality to some extent, CoPDT demonstrates substantially greater robustness. This highlights the superior generalization capability of the constraint prioritized prompt encoder in handling unseen prompts.

Table 11: Detailed results of the ablation studies done in the Multi-constraint Multi-budget setting.

| Task | Ours (MT) | | W/o CP | | Det CP | | W/o CD | | W/o PE | | Simp PE | | Small DT | |
|---|---|---|---|---|---|---|---|---|---|---|---|---|---|---|
| | r↑ | c↓ | r↑ | c↓ | r↑ | c↓ | r↑ | c↓ | r↑ | c↓ | r↑ | c↓ | r↑ | c↓ |
| PointButton1 | **0.04** | **0.55** | 0.49 | 3.94 | **0.03** | **0.62** | 0.09 | 1.14 | **0.08** | **0.94** | 0.02 | 0.60 | 0.07 | 0.73 |
| PointButton2 | **0.08** | **0.98** | 0.32 | 3.31 | **0.03** | **0.95** | 0.11 | 1.41 | 0.07 | 1.01 | 0.10 | 1.21 | 0.11 | 1.19 |
| PointCircle1 | 0.55 | 1.09 | **0.57** | **0.93** | 0.52 | 1.05 | 0.55 | 1.05 | **0.48** | **0.70** | 0.53 | 0.86 | 0.53 | 0.71 |
| PointCircle2 | 0.57 | 1.75 | 0.60 | 1.92 | 0.55 | 1.85 | 0.57 | 1.85 | 0.57 | 2.59 | 0.55 | 1.51 | 0.53 | 1.46 |
| PointGoal1 | **0.24** | **0.30** | 0.62 | 1.44 | **0.20** | **0.35** | 0.33 | 0.53 | 0.22 | 0.32 | 0.26 | 0.36 | 0.28 | 0.48 |
| PointGoal2 | **0.26** | **0.66** | 0.49 | 2.33 | **0.23** | **0.63** | 0.30 | 0.89 | 0.22 | 0.79 | 0.22 | 0.77 | 0.27 | 0.99 |
| PointPush1 | **0.12** | **0.69** | 0.25 | 1.40 | **0.10** | **0.57** | 0.17 | 0.7 | 0.13 | 0.52 | 0.12 | 0.56 | 0.17 | 0.69 |
| PointPush2 | **0.11** | **0.83** | 0.15 | 1.25 | **0.10** | **0.78** | 0.10 | 0.94 | 0.11 | 0.77 | 0.11 | 0.80 | 0.14 | 1.20 |
| CarButton1 | **0.04** | **0.89** | 0.23 | 4.70 | **0.02** | **0.66** | 0.07 | 0.77 | 0.02 | 0.65 | 0.04 | 0.67 | 0.02 | 0.60 |
| CarButton2 | **-0.02** | **0.94** | 0.17 | 4.77 | **-0.02** | **0.77** | -0.05 | 1.56 | 0.01 | 1.07 | -0.01 | 1.05 | -0.04 | 1.14 |
| CarCircle1 | 0.50 | 2.89 | 0.52 | 4.10 | 0.49 | 2.87 | 0.50 | 3.17 | 0.42 | 2.44 | 0.51 | 3.27 | 0.56 | 4.53 |
| CarCircle2 | 0.34 | 1.67 | 0.55 | 4.61 | 0.31 | 1.40 | 0.35 | 2.05 | 0.46 | 4.06 | 0.32 | 1.37 | 0.38 | 2.41 |
| CarGoal1 | **0.22** | **0.32** | 0.50 | 1.32 | **0.19** | **0.29** | 0.24 | 0.37 | 0.19 | 0.28 | 0.21 | 0.34 | 0.26 | 0.45 |
| CarGoal2 | **0.13** | **0.91** | 0.32 | 1.83 | **0.15** | **0.94** | 0.20 | 1.01 | **0.15** | **0.76** | 0.14 | 0.94 | 0.19 | 1.06 |
| CarPush1 | **0.18** | **0.48** | 0.24 | 0.73 | **0.16** | **0.43** | 0.20 | 0.45 | 0.17 | 0.33 | 0.18 | 0.35 | 0.19 | 0.38 |
| CarPush2 | **0.06** | **0.62** | 0.16 | 2.25 | **0.04** | **0.46** | 0.07 | 1.15 | 0.04 | 0.54 | 0.05 | 0.61 | 0.06 | 0.81 |
| SwimmerVelocityV0 | **0.69** | **0.84** | 0.72 | 0.72 | 0.53 | 0.64 | 0.72 | 0.82 | 0.67 | 5.79 | 0.70 | 1.93 | 0.67 | 3.12 |
| SwimmerVelocityV1 | **0.61** | **0.74** | 0.66 | 0.65 | 0.43 | 1.38 | 0.66 | 0.69 | 0.56 | 0.51 | 0.59 | 0.74 | 0.56 | 0.76 |
| HopperVelocityV0 | 0.57 | 4.28 | 0.83 | 2.31 | 0.41 | 3.77 | 0.60 | 4.13 | 0.55 | 2.69 | 0.55 | 3.25 | 0.37 | 1.85 |
| HopperVelocityV1 | 0.27 | 1.09 | 0.64 | 3.53 | **0.23** | **0.96** | 0.37 | 1.60 | **0.44** | **0.52** | 0.28 | 1.43 | 0.16 | 2.08 |
| HalfCheetahVelocityV0 | **0.70** | **0.36** | 0.95 | 0.78 | 0.67 | 0.31 | 0.76 | 0.40 | 0.75 | 13.77 | **0.7** | **0.38** | 0.78 | 0.17 |
| HalfCheetahVelocityV1 | 0.75 | 1.22 | **0.95** | **0.27** | 0.73 | 1.06 | 0.80 | 1.13 | **0.73** | **0.65** | 0.88 | 0.49 | 0.67 | 0.69 |
| Walker2dVelocityV0 | 0.35 | 4.44 | 0.28 | 2.47 | 0.36 | 4.51 | 0.36 | 4.49 | 1.44 | 27.45 | 0.40 | 5.57 | 0.36 | 4.64 |
| Walker2dVelocityV1 | **0.66** | **0.73** | 0.74 | 0.09 | **0.66** | **0.72** | 0.67 | 0.72 | 0.69 | 0.71 | 0.72 | 0.59 | 0.58 | 2.28 |
| AntVelocityV0 | 0.95 | 4.89 | 0.94 | 1.65 | 0.96 | 5.01 | 0.96 | 4.99 | 0.85 | 7.02 | 0.98 | 8.45 | 0.99 | 10.13 |
| AntVelocityV1 | 0.92 | 3.42 | **0.98** | **0.71** | 0.91 | 3.80 | 0.95 | 3.13 | 0.92 | 5.24 | 0.96 | 3.44 | 0.96 | 3.29 |
| Average | 0.38 | 1.45 | 0.53 | 2.08 | 0.35 | 1.42 | 0.41 | 1.58 | 0.42 | 3.16 | 0.39 | 1.60 | 0.38 | 1.84 |

## F.14 Detailed main ablation results

In this section, we also provide the detailed results of the ablation studies, as shown in Table 11. Here, we provide an additional baseline **Det CP**, which models $q_\phi$ deterministically. The results show that Det CP achieves improved safety performance, but this comes at the cost of lower reward performance. Moreover, due to its deterministic nature, Det CP is unable to adjust conservatism through $\beta_{end}$, lacking flexibility.

## F.15 Standard deviation of the main results

In this section, we present the standard deviation of CoPDT and various baselines three different random seeds, as shown in Table 12. The results demonstrate that CoPDT consistently achieves significantly lower standard deviation compared to other baselines, indicating its superior stability.

## G More details about related work

**Safe RL** Safe RL is a kind of machine learning approach aimed at learning policies that maximize cumulative rewards while adhering to additional predefined safety constraints (Gu et al., 2022). Safe RL algorithms are broadly divided into two categories: safe exploration and safe optimization (García & Fernández, 2015). Safe exploration algorithms do not focus on directly optimizing the policy. Instead, they aim to modify the policy's behavior through additional mechanisms to prevent violations of safety constraints. A typical example of these algorithms is shielding-based methods (Alshiekh et al., 2018; Cheng et al., 2019; Xiao et al., 2023), which construct or learn logical structures known as "shields" or "barriers" that ensure the actions taken in a given state comply with the safety constraints. However, the decoupling from policy learning of safe exploration methods results in lower learning efficiency, leading to a growing focus on safe optimization algorithms. Safe

Table 12: Detailed standard deviation results of the main experiment. All results are computed using three different random seeds. For experiments involving multiple budgets, the reported standard deviation is averaged across all budgets.

| Task | Single-constraint Multi-budget | | | | | | | | Multi-constraint Single-budget | | | | | | Multi-constraint Multi-budget | | | | | |
|---|---|---|---|---|---|---|---|---|---|---|---|---|---|---|---|---|---|---|---|---|
| | CPQ | | CDT | | FISOR | | CoPDT | | MTCDT | | Prompt-CDT | | CoPDT | | MTCDT | | Prompt-CDT | | CoPDT | |
| | r_std | c_std | r_std | c_std | r_std | c_std | r_std | c_std | r_std | c_std | r_std | c_std | r_std | c_std | r_std | c_std | r_std | c_std | r_std | c_std |
| PointButton1 | 0.05 | 0.30 | 0.03 | 0.48 | 0.03 | 1.32 | 0.01 | 0.26 | 0.07 | 1.03 | 0.01 | 0.97 | 0.03 | 0.22 | 0.06 | 1.11 | 0.03 | 0.93 | 0.04 | 0.27 |
| PointButton2 | 0.05 | 1.23 | 0.04 | 0.82 | 0.03 | 1.10 | 0.04 | 0.49 | 0.07 | 0.20 | 0.01 | 0.80 | 0.06 | 0.34 | 0.07 | 0.78 | 0.04 | 0.96 | 0.05 | 0.33 |
| PointCircle1 | 0.13 | 1.03 | 0.01 | 0.21 | 0.08 | 4.73 | 0.01 | 0.31 | 0.01 | 0.15 | 0.00 | 0.22 | 0.01 | 0.27 | 0.02 | 0.13 | 0.01 | 0.18 | 0.01 | 0.20 |
| PointCircle2 | 0.24 | 5.19 | 0.01 | 0.17 | 0.04 | 4.91 | 0.01 | 0.12 | 0.01 | 0.47 | 0.01 | 0.21 | 0.03 | 0.81 | 0.01 | 0.29 | 0.02 | 0.51 | 0.03 | 0.59 |
| PointGoal1 | 0.10 | 0.20 | 0.02 | 0.15 | 0.01 | 1.56 | 0.03 | 0.08 | 0.05 | 0.24 | 0.02 | 0.23 | 0.04 | 0.15 | 0.06 | 0.29 | 0.03 | 0.38 | 0.04 | 0.12 |
| PointGoal2 | 0.14 | 1.40 | 0.05 | 0.44 | 0.06 | 1.04 | 0.01 | 0.05 | 0.04 | 0.49 | 0.04 | 0.41 | 0.03 | 0.05 | 0.04 | 0.52 | 0.04 | 0.87 | 0.02 | 0.20 |
| PointPush1 | 0.06 | 0.77 | 0.03 | 0.25 | 0.05 | 0.74 | 0.01 | 0.15 | 0.02 | 0.30 | 0.03 | 0.25 | 0.02 | 0.20 | 0.02 | 0.18 | 0.02 | 0.41 | 0.03 | 0.12 |
| PointPush2 | 0.08 | 0.77 | 0.03 | 0.32 | 0.04 | 0.87 | 0.01 | 0.59 | 0.03 | 0.66 | 0.05 | 0.40 | 0.01 | 0.50 | 0.04 | 0.62 | 0.06 | 0.44 | 0.01 | 0.30 |
| CarButton1 | 0.06 | 2.13 | 0.03 | 0.46 | 0.05 | 0.12 | 0.00 | 0.27 | 0.03 | 0.54 | 0.07 | 0.47 | 0.02 | 0.29 | 0.05 | 1.15 | 0.05 | 0.98 | 0.01 | 0.24 |
| CarButton2 | 0.10 | 1.94 | 0.05 | 0.53 | 0.01 | 0.34 | 0.01 | 0.39 | 0.05 | 0.60 | 0.02 | 0.27 | 0.02 | 0.37 | 0.04 | 1.15 | 0.05 | 0.82 | 0.01 | 0.32 |
| CarCircle1 | 0.09 | 2.38 | 0.04 | 0.65 | 0.03 | 4.30 | 0.05 | 1.09 | 0.01 | 0.22 | 0.03 | 0.57 | 0.00 | 0.13 | 0.01 | 0.28 | 0.02 | 0.78 | 0.02 | 0.19 |
| CarCircle2 | 0.04 | 0.99 | 0.02 | 0.68 | 0.03 | 4.28 | 0.02 | 0.23 | 0.01 | 0.69 | 0.01 | 0.77 | 0.04 | 0.15 | 0.02 | 0.69 | 0.03 | 1.15 | 0.02 | 0.29 |
| CarGoal1 | 0.04 | 0.39 | 0.03 | 0.26 | 0.03 | 0.57 | 0.04 | 0.04 | 0.04 | 0.30 | 0.05 | 0.43 | 0.05 | 0.03 | 0.03 | 0.37 | 0.05 | 0.52 | 0.05 | 0.10 |
| CarGoal2 | 0.07 | 1.60 | 0.04 | 0.57 | 0.03 | 0.33 | 0.04 | 0.13 | 0.06 | 0.51 | 0.04 | 0.45 | 0.01 | 0.11 | 0.06 | 0.51 | 0.04 | 0.47 | 0.03 | 0.25 |
| CarPush1 | 0.16 | 4.97 | 0.02 | 0.21 | 0.02 | 0.46 | 0.02 | 0.07 | 0.01 | 0.56 | 0.03 | 0.17 | 0.02 | 0.34 | 0.02 | 0.31 | 0.02 | 0.24 | 0.01 | 0.38 |
| CarPush2 | 0.07 | 2.06 | 0.03 | 0.82 | 0.05 | 0.71 | 0.01 | 0.11 | 0.02 | 0.52 | 0.04 | 0.47 | 0.01 | 0.36 | 0.03 | 0.67 | 0.04 | 0.30 | 0.03 | 0.40 |
| SwimmerVelocityV0 | 0.07 | 0.72 | 0.01 | 0.33 | 0.01 | 0.17 | 0.03 | 0.18 | 0.02 | 2.29 | 0.00 | 0.06 | 0.01 | 0.04 | 0.02 | 3.07 | 0.00 | 0.06 | 0.01 | 0.06 |
| SwimmerVelocityV1 | 0.09 | 1.02 | 0.02 | 0.29 | 0.03 | 0.20 | 0.09 | 0.12 | 0.02 | 0.20 | 0.01 | 0.39 | 0.02 | 0.20 | 0.01 | 0.16 | 0.01 | 0.36 | 0.03 | 0.22 |
| HopperVelocityV0 | 0.05 | 1.59 | 0.05 | 0.11 | 0.13 | 0.44 | 0.04 | 0.45 | 0.23 | 8.33 | 0.05 | 2.35 | 0.09 | 3.32 | 0.20 | 8.17 | 0.03 | 2.66 | 0.11 | 2.24 |
| HopperVelocityV1 | 0.14 | 2.05 | 0.04 | 1.39 | 0.08 | 0.65 | 0.03 | 0.15 | 0.01 | 1.53 | 0.06 | 5.32 | 0.01 | 0.26 | 0.04 | 1.46 | 0.07 | 3.17 | 0.09 | 0.29 |
| HalfCheetahVelocityV0 | 0.34 | 1.55 | 0.01 | 0.37 | 0.00 | 0.00 | 0.02 | 0.04 | 0.14 | 12.49 | 0.01 | 0.88 | 0.16 | 0.56 | 0.13 | 13.14 | 0.01 | 4.91 | 0.08 | 0.27 |
| HalfCheetahVelocityV1 | 0.21 | 1.56 | 0.00 | 0.15 | 0.00 | 0.00 | 0.07 | 0.04 | 0.03 | 0.65 | 0.02 | 0.33 | 0.09 | 0.27 | 0.04 | 0.75 | 0.02 | 0.36 | 0.08 | 0.44 |
| Walker2dVelocityV0 | 0.03 | 0.46 | 0.03 | 0.59 | 0.03 | 0.79 | 0.02 | 0.75 | 0.38 | 8.23 | 0.42 | 9.43 | 0.01 | 0.10 | 0.37 | 9.27 | 0.38 | 8.28 | 0.02 | 0.47 |
| Walker2dVelocityV1 | 0.02 | 0.25 | 0.01 | 0.06 | 0.12 | 0.59 | 0.02 | 0.07 | 0.01 | 0.20 | 0.03 | 0.20 | 0.07 | 0.61 | 0.01 | 0.23 | 0.06 | 0.13 | 0.05 | 0.39 |
| AntVelocityV0 | 0.00 | 0.00 | 0.00 | 0.09 | 0.01 | 0.00 | 0.01 | 0.14 | 0.00 | 0.02 | 0.00 | 0.51 | 0.02 | 1.97 | 0.00 | 0.07 | 0.01 | 0.43 | 0.02 | 1.59 |
| AntVelocityV1 | 0.00 | 0.00 | 0.01 | 0.13 | 0.01 | 0.00 | 0.01 | 0.69 | 0.00 | 0.12 | 0.00 | 0.06 | 0.01 | 1.25 | 0.00 | 0.11 | 0.01 | 0.12 | 0.05 | 0.39 |
| Average | 0.09 | 1.41 | 0.03 | 0.41 | 0.04 | 1.16 | 0.02 | 0.27 | 0.05 | 1.60 | 0.04 | 1.02 | 0.04 | 0.50 | 0.05 | 1.75 | 0.04 | 1.17 | 0.04 | 0.42 |

optimization algorithms typically model the problem as a CMDP, with Lagrangian multiplier-based algorithms being the mainstream solution, as discussed in the main paper. Other than Lagrangian multiplier-based algorithms, trust region methods are among the most prevalent approaches in safe optimization. They attempt to keep policies within a safe trust region during updates via low-order Taylor expansions (Achiam et al., 2017) or variational inference (Liu et al., 2022). Due to their robust learning process, trust region methods are also further applied to multi-agent scenarios (Gu et al., 2023). However, their on-policy nature results in lower data efficiency. In response, recent works have increasingly focused on off-policy safe optimization. CAL (Wu et al., 2024) improves the optimization of Lagrange multipliers using the augmented Lagrangian method and enhances the conservatism of the cost function learned off-policy via the use of upper confidence bound. Meanwhile, SafeDreamer (Huang et al., 2024) increases data efficiency by learning an environment model and using model rollouts for data augmentation. Recently, more attention has been directed toward safety-conditioned RL. CCPO (Yao et al., 2023) effectively adapts to different safety thresholds in an online algorithm by incorporating the safety threshold into the input of CVPO. On the other hand, SDT (Guo et al., 2024) attempts to integrate safety prior knowledge expressed through temporal logic into the input of DT, further enhancing the policy's safety performance while adapting to various temporal logic safety constraints. This provides a fresh perspective for the practical application of safety RL. With the growing body of research on safe RL, these algorithms have found increasing applications in various fields. Notable examples include ensuring the safety of vehicles in autonomous driving (Kiran et al., 2021) and safeguarding robots in industrial settings (Brunke et al., 2022).

**Offline RL**   Offline RL trains policies using pre-collected datasets, avoiding real-world trial and error, which is critical for deploying RL in practical settings. Its primary challenge is addressing the extrapolation errors (Prudencio et al., 2023). Methods like BCQ (Fujimoto et al., 2019) and CQL (Kumar et al., 2020) tackle this by constraining actions to those seen in the offline data or by penalizing unseen actions. Others, such as MOReL (Kidambi et al., 2020) and MOPO (Yu et al., 2020), learn the environment models from the offline data and utilize these models with uncertainty estimates to avoid OOD regions with low model accuracy. However, these methods only focus on reducing extrapolation errors, without addressing the challenge of generalizing in OOD areas. To tackle this limitation, MOREC (Luo et al., 2024) employs adversarial learning in the model learning process, improving model generalization abilities.

**Meta RL**    Meta RL is similar to multi-task RL, with both involving multi-task training. However, Meta RL does not receive additional expert trajectories as prompts during testing. Instead, it must collect data in the unknown environment to generate prompts. Additionally, it focuses on training across large-scale similar tasks for generalization to new ones (Zhu et al., 2023). PEARL (Rakelly et al., 2019) uses a probabilistic encoder to facilitate task identification and employs Thompson sampling for data collection in new environments. Other works, like FOCAL (Li et al., 2021) and CORRO (Yuan & Lu, 2022), focus on designing contrastive loss functions for the encoder, improving task encoding robustness. COSTA (Guan et al., 2024) first considers safety in meta RL, designing a cost-based contrastive loss and a safety-aware data collection framework, improving policy safety in both task identification and deployment.

## H    Discussion on limitations

CoPDT does indeed have some limitations. Firstly, CoPDT does not exhibit strong generalization capabilities, which is actually anticipated. One reason is that in our multi-task training, we did not select a large set of similar tasks (such as numerous variations of velocity) but rather chose a variety of tasks with significant differences. We believe this scenario is more common in practice. In contrast, past meta-RL methods, which emphasize generalization, typically demonstrate limited generalization ability only when trained on a large number (dozens or even hundreds) of similar tasks. Another reason is that generalizing to unseen safety constraints has limited practical value. Safety constraints are stringent, and in real-world applications, there is often a preference for retraining or fine-tuning a model to obtain a safer policy rather than relying on an unprovable generalized policy. Secondly, CoPDT may introduce larger errors in RTG modeling for tasks with larger and more diverse reward distributions (e.g., velocity tasks), leading to a drop in performance compared to tasks with smaller and more concentrated reward distributions (e.g., point, car tasks). Therefore, further research into how to model RTG effectively for tasks with wide-ranging reward distributions is needed. Finally, CoPDT uses different input-output heads for tasks in different environments. As a result, when fine-tuning in unseen environments, it requires the re-initialization of additional input-output heads, which may reduce the efficiency of knowledge transfer. Thus, how to facilitate knowledge sharing and transfer across tasks in different environments (with different state action spaces, and transition functions) remains an important avenue for future research.

## I    Broader societal impacts

The goal of the work presented in this paper is to advance the development of sequence modeling-based methods in offline safe RL. These methods, built on the Transformer, are intended to more effectively scale to multi-task scenarios and beyond, laying a foundation for future research on scaling policies in RL and safe RL. In the future, our approach holds promise for broader application across a variety of decision-making scenarios involving budget constraints. Meanwhile, we believe our work does not pose any foreseeable negative societal impacts.

