# OpenReview forum: "Adaptable Safe Policy Learning from Multi-task Data with Constraint Prioritized Decision Transformer"
_NeurIPS.cc/2025/Conference — NeurIPS 2025 poster_

### Official Review · Reviewer_gaMd · 2025-06-24

**Clarity:** 3
**Significance:** 3
**Originality:** 3
**Rating:** 4
**Confidence:** 4

**Summary:**

This paper tackles the challenge in safe reinforcement learning: how to learn policies from offline data that can adapt to different safety requirements across multiple tasks. The authors focus on scenarios where an agent must handle various safety constraints (like different speed limits for different robots) and adapt to changing safety budgets (how much violation is tolerable) without direct environment interaction. The key insight is that existing decision transformer approaches struggle when faced with multiple constraints because they rely on manually specified reward targets (Return-To-Go) that often conflict with safety requirements (Cost-To-Go). When these conflict, the agent tends to prioritize rewards over safety, leading to constraint violations.

**Questions:**

- The results show constraint violations in zero-shot settings (e.g., HopperV2 with 9.96±3.06 cost vs 1.0 budget). This suggests the prompt encoder is memorizing specific constraint patterns rather than learning generalizable safety concepts. Have you investigated alternative architectures that could learn more compositional constraint representations? For example, using contrastive learning objectives that explicitly maximize distances between different constraint types while maintaining invariance to trajectory-specific features?

- Could you please provide analysis on what the prompt encoder actually learns? Specifically, visualization of prompt embedding gradients with respect to different trajectory features might reveal whether it's capturing constraint semantics or trajectory statistics.

- Would it be possible to integrate CoPDT with existing safety verification frameworks? For example, could the learned constraint representations be used to define control barrier functions?

- Is it possible to derive PAC-style bounds on constraint violation probability given finite offline data?

**Ethical Concerns:**

["NO or VERY MINOR ethics concerns only"]

**Limitations:**

-  The fundamental limitation to binary costs is not discussed, despite many real-world safety metrics being naturally continuous

- Should explicitly state that CoPDT provides no formal safety certificates or probabilistic bounds, making it unsuitable for truly safety-critical applications without additional safeguards

Constructive Suggestions:

- Would it be possible to add a "When NOT to use CoPDT" section explicitly stating: systems requiring formal guarantees, continuous cost functions, strict real-time constraints, or high-stakes deployment without extensive testing

**Paper Formatting Concerns:**

NA.

**Quality:**

3

**Strengths And Weaknesses:**

Technical Strengths:

- The constraint prioritized prompt encoder leverages the binary nature of costs to create separate neural pathways for safe and unsafe trajectory segments, amplifying the sparse cost signal that would otherwise be dominated by dense state-action information.
- Rather than relying on fixed Return-To-Go values, the CPRTG mechanism learns a probabilistic model of appropriate reward targets conditioned on current safety budgets, solving the conflict between reward maximization and constraint satisfaction in decision transformers.
-The prompt encoder maintains discriminative power even when evaluated on out-of-distribution trajectories (Medium1/Medium2 prompts), suggesting learned representations capture fundamental constraint properties rather than superficial trajectory statistics.

Technical Weaknesses:

- The approach fails to develop robust constraint representations that transfer to unseen cost functions - the learned prompt embeddings are essentially memorizing specific constraint patterns rather than learning generalizable safety concepts.

- The CPRTG generator struggles with high-variance reward distributions, as evidenced by velocity tasks showing systematically worse safety performance than navigation tasks with concentrated rewards.

- The architecture cannot incrementally learn new constraints without retraining from scratch, as the prompt encoder's fixed dimensionality and the environment-specific components prevent online adaptation.

---

> ### Author Rebuttal · Authors · 2025-07-31
>
> Thank you for your insightful reviews. We appreciate your feedback and hope that our response addresses your concerns.
>
> ### Q1 More analysis on zero-shot generalization
>
> Currently, for each task type, CoPDT is pre-trained using only 2 similar tasks. We acknowledge that this limited diversity may restrict the model’s ability to achieve stronger zero-shot generalization performance. **However, this limitation can be mitigated by pre-training on a broader set of similar tasks**.
>
> Therefore, we conducted **an additional experiment where we trained policies under a larger number of velocity thresholds** ($2 \to 6$) in the AntVelocity and HopperVelocity environments. The goal is to examine whether exposing CoPDT to more similar tasks during training can improve its zero-shot generalization ability. Please refer to our response to **Q5 of Reviewer YyH8** for detailed experimental settings and results.
>
> It can be observed that when CoPDT is pre-trained on a larger number of similar tasks—similar to the setup in meta-RL—the model exhibits **significant improvements in safety performance** on both interpolation (V2) and extrapolation (V7) generalization tasks. The improvement is particularly notable in the Hopper environment, which may be attributed to the large threshold gap between the original two Hopper tasks, making pre-training on only two tasks insufficient. These results support the claim that our **constraint-prioritized prompt encoder not only captures task-specific memory but also possesses a certain degree of generalization capability to unseen constraints**.
>
> ### Q2 CPRTG generator with high-variance reward distributions
>
> We acknowledge that this reflects a current limitation of the CPRTG generator under the **Gaussian modeling assumption**. However, we believe that **adopting more powerful generative models with strong multi-modal distribution modeling capabilities**—such as diffusion models or flow matching—offers a promising solution. Enhancing the CPRTG generator in this way is an important direction for future research.
>
> ### Q3 Incrementally learn new constraints
>
> We would like to clarify that **CoPDT is inherently capable of incrementally adapting to new constraints and environments**.
>
> In the multi-constraint setting, both the prompt encoder and the DT policy in CoPDT adopt a three-layer architecture: the bottom and top layers are environment-specific encoders and decoders, designed to handle different state and action space dimensions across environments, as stated in **Sec. 3.1**. **The middle layer, which forms the core of both the prompt encoder and DT policy, is shared across all tasks and environments to enable effective knowledge transfer.**
>
> As a result, when encountering new tasks with previously unseen state and action spaces, we can train new encoders and decoders for those tasks while **reusing the shared middle layers** of the prompt encoder and DT policy. This design naturally supports incremental adaptation without retraining the entire model from scratch.
>
> We have evaluated this capability in an experimental setup described in **App. F.7 (Fig. 12)**, where we perform few-shot transfer to unseen robots whose state and action spaces were not present during training. The results show that CoPDT with full fine-tuning achieves better safety performance compared to learning from scratch, even in these entirely new environments—demonstrating the effectiveness of multi-task training and knowledge sharing.
>
> ### Q4 What encoder learns?
>
> Actually, our prompt encoder is **capable of effectively extracting and representing information from the cost signal**.
>
> Since the cost signal in our prompt encoder is **not used to compute gradients**, but instead serves as a **switch to select between sub-networks**, we are unable to provide traditional gradient-based interpretation methods. However, we propose an alternative and more intuitive visualization approach: **directly comparing the outputs of the same samples (s,a,r,s’) processed through $p_s$ and $p_u$**.
>
> To do this, we collected 5,000 transition samples in PointCircle1 and passed them through both $p_s$ and $p_u$, obtaining two sets of 5,000 embedding vectors. While we are unable to include t-SNE visualizations of these vectors due to rebuttal limitations, we compute the **cosine similarity** between the mean vectors of the two sets, which yields a value of **0.415**. This relatively low similarity indicates that the encodings produced by $p_s$ and $p_u$ differ substantially, confirming that **the selection mechanism based on cost information has a significant effect on the encoding**.
>
> These findings provide empirical support for the effectiveness of our constraint prioritized prompt encoder, showing that the cost signal meaningfully influences the learned task representation, even without direct gradient supervision.
>
> ### Q5 Integrate CoPDT with existing safety verification frameworks
>
> Indeed, the constraint representations learned in CoPDT can be **naturally integrated with existing safety verification frameworks**, such as control barrier functions (CBFs).
>
> In the case of neural-network-based CBFs, the constraint representation can be directly treated as part of the state and used as an additional input to the neural network, enabling task-adaptive safety filtering.
>
> For non-parametric CBFs, one can construct a set of prototype task embeddings by generating constraint representations from a small number of trajectories for each training task. During testing, a trajectory from the target task can be used to compute its constraint representation, which is then matched to the most similar prototype. The corresponding CBF of the matched prototype task can then be applied to enforce safety in the target task.
>
> ### Q6 PAC-style bounds on constraint violation probability？
>
> Since CoPDT is trained using the behavior cloning objective from DT, **it does not naturally lend itself to PAC-style theoretical guarantees** as commonly found in traditional TD-learning frameworks.
>
> To complement this limitation, we provide empirical evidence in App. A by reporting **the performance gap between the learned policy and the behavior policy in the offline dataset**. These results highlight the importance of using the CPRTG generator to better align the RTG with the offline data distribution, thereby mitigating distributional shift and improving policy performance.
>
> ### Q7 Adding a "When NOT to use CoPDT” section
>
> Thank you very much for your valuable suggestion. We will incorporate this section in the next version of the paper, explicitly stating:
>
> > "Systems that require formal safety guarantees, continuous cost functions, strict real-time constraints, or high-stakes deployment without extensive testing are not suitable for deploying CoPDT."
> >
>
> This clarification will help better delineate the scope and applicability of our method.

---

### Official Review · Reviewer_YyH8 · 2025-06-30

**Clarity:** 3
**Significance:** 2
**Originality:** 2
**Rating:** 4
**Confidence:** 3

**Summary:**

This work proposes a novel offline safe reinforcement learning (RL) framework built upon the Decision Transformer (DT) architecture, designated as Constraint-Prioritized Decision Transformer (CoPDT). The proposed framework is designed to tackle the limited adaptability of existing DT-based approaches in multi-task scenarios characterized by diverse constraints and dynamic safety budgets. CoPDT encompasses two pivotal innovations: firstly, a constraint-prioritized prompt encoder that utilizes sparse binary cost signals to precisely identify task-specific constraints; secondly, a Constraint-Prioritized Return-To-Go (CPRTG) mechanism, which dynamically generates return-to-go targets based on Cost-To-Go (CTG) and safety budgets, thus reconciling the conflict between reward maximization and adherence to safety requirements. Comprehensive experiments conducted on 26 tasks from the OSRL benchmark validate that CoPDT retains competitive reward performance while demonstrating its efficacy in multi-constraint and multi-budget settings.

**Questions:**

1.The constraint prioritized prompt encoder relies on binary cost signals (c=0/1), yet real-world data often contains noisy or ambiguous cost labels. How does CoPDT perform under noisy cost signals? A robustness analysis would enhance confidence in its practical application, and improved performance in noisy environments would elevate the quality score.

2.CPRTG utilizes CTG-based β decay to balance conservatism, with β_start and β_end being fixed. How sensitive is CoPDT to these hyperparameters across different tasks? A sensitivity analysis would clarify their impact on the risk-reward trade-off, and stable performance across varying values would strengthen the reliability of the method.

3.This paper focuses on the offline setting, but many practical scenarios require online adaptation. Can CoPDT be extended to online fine-tuning (e.g., with limited environmental interactions)? A preliminary evaluation of online adaptability would enhance its significance, as successful results would broaden its applicability.

4.The capability of zero-shot generalization to unseen constraints is limited (Fig. 4a). Could the integration of meta-learning (e.g., learning constraint embeddings across more diverse tasks) improve this? Evidence that meta-learning enhances zero-shot performance would address a key limitation and improve the overall score.

**Ethical Concerns:**

["NO or VERY MINOR ethics concerns only"]

**Limitations:**

yes

**Quality:**

3

**Strengths And Weaknesses:**

Strengths：

1. The method proposed in this paper is strongly motivated, with both the constraint-prioritized fast encoder and the CPRTG mechanism embodying clear design principles.

2. In terms of experiments, it covers a variety of tasks, conducts rigorous comparisons with baseline methods (CPQ, CDT, FISOR), and verifies the contributions of core components (fast encoder, CPRTG) through ablation experiments.

3. This paper skillfully combines existing methods to address the challenges of offline reinforcement learning in multi-task and multi-constraint scenarios. The modular design of CoPDT and sufficient empirical evidence provide some insights for relevant researchers.

Weaknesses：

1. Both the immediate encoding and inverse dynamics model presented in this paper are existing algorithms, and the core method of this paper is a combination of such existing methods, which significantly diminishes the innovativeness of this work.

2. The constraint-prioritized immediate encoding strategy relies on accurate cost estimation, and deviations in cost estimation appear to significantly impact model performance.

3.A clearer exposition of how β decay in CTG balances safety and reward gains would facilitate readers' understanding of the method proposed in this paper.

---

> ### Author Rebuttal · Authors · 2025-07-31
>
> Thank you for your insightful reviews. We appreciate your feedback and hope that our response addresses your concerns.
>
> ### Q1 Novelty?
>
> CoPDT tackles the **unique challenges of combining offline safe RL with multi-task sequence modeling**, introducing the following key innovations:
>
> - **Unified modeling for multi-task offline safe RL:** CoPDT is the first to explicitly learn a unified model adaptable to tasks with varying constraints and budgets. This improves offline data efficiency and few-shot transfer to new constraints in limited-data scenarios.
> - **Resolving reward-safety conflicts in multi-budget settings:** In such settings, optimizing for both high reward and strict safety often leads to conflicts that limit safety performance. To address this, CoPDT introduces the CPRTG token, conditioning RTG generation on CTG to prioritize safety while encouraging maximal feasible rewards. Experiments demonstrate that CoPDT achieves significantly better safety performance than CDT and other baselines under multi-budget conditions, validating the effectiveness of this approach. To our knowledge, no prior work has addressed this problem from the perspective of prioritizing CTG during RTG generation, making this a novel solution strategy.
> - **Constraint prioritized task representation for multi-constraint adaptation:** Inspired by hyper-networks, we propose a constraint prioritized prompt encoder explicitly integrates binary cost signals into encoder structure design for more effective constraint representation. As shown in **Fig. 3(b) of Sec. 4.3** and the experiments in **App. F.6**, this encoder outperforms standard MLP-based encoders, especially when prompt quality is low—demonstrating stronger robustness in constraint representation.
>
> In summary, CoPDT introduces **key innovations in CPRTG modeling and structured prompt encoding**, offering a **targeted solution to multi-task offline safe RL** rather than merely combining existing techniques.
>
> ### Q2 Noisy or ambiguous cost signals
>
> In line with most prior work in offline safe RL, we focus on settings with clear, binary cost signals. However, we acknowledge that dealing with noisy or ambiguous cost signals is an important and practical challenge, and is valuable for future works.
>
> To address this, we assume access to a **language-based descriptions** **for the cost-related rules.** Under this setting, we believe a promising approach is to **apply the language-guided reward generation methods [1,2] to cost modeling, and integrate them with CoPDT**. Specifically, cost-related rule descriptions, along with observation and other relevant information, are input into a LLM, which generates a cost function aligned with the described rules. This function can then be verified and refined via human inspection or a few labeled samples, and used to annotate datasets with noisy or ambiguous cost labels.
>
> We conducted preliminary experiments in this direction, discarding the original (potentially noisy) cost labels and using GPT-4o-mini to generate a new, slightly more conservative cost function ( to mitigate the impact of ambiguity or imprecision in the rule-based descriptions) based on the language description. As shown in the table below, CoPDT trained on LLM-labeled data maintains strong safety performance—**even surpassing results obtained using the original ground-truth labels in terms of safety**. These findings validate the effectiveness of our proposed approach in scenarios with noisy or even absent cost labels.
>
> | Task | LLM-labled CoPDT |
> | --- | --- |
> |  | r↑  c↓ |
> | PB1 | **-0.01, 0.26** |
> | PB2 | 0.08, 1.02 |
> | PC1 | **0.33, 0.00** |
> | PC2 | **0.49, 0.45** |
> | PG1 | **0.20, 0.36** |
> | PG2 | **0.21, 0.64** |
> | PP1 | **0.11, 0.39** |
> | PP2 | **0.09, 0.56** |
> | CB1 | **0.00, 0.35** |
> | CB2 | **-0.03, 0.63** |
> | CC1 | 0.44, 2.60 |
> | CC2 | 0.40, 2.26 |
> | CG1 | **0.19, 0.32** |
> | CG2 | **0.11, 0.80** |
> | CP1 | **0.15, 0.38** |
> | CP2 | **0.07, 0.91** |
> | Average | 0.18, 0.75 |
>
> In the absence of language-based cost rule descriptions, we advocate exploring **robust algorithmic designs** and shifting to an **online training setup**. In offline settings with potentially erroneous cost labels, it is difficult to reliably distinguish trustworthy from corrupted data. In contrast, under an online setting, one can **leverage existing safe and robust reinforcement learning frameworks by introducing an adversary that perturbs the cost labels of online samples with a certain probability** [3]. This enables learning policies that are both robust to label noise and maintain strong safety—representing a promising direction for future research.
>
> ### Q3 Sensitivity analysis on $\beta-$decay
>
> Different values of $\beta_{\text{end}}$ do affect reward and cost, but **the impact on cost remains within 0.1**, indicating minimal influence on policy stability. Instead, it offers CoPDT a useful mechanism for **fine-tuning performance during deployment**.
>
> We agree that that studying the impact of $\beta$-decay on the risk-reward trade-off is important. To this end, we conduct a sensitivity analysis of $\beta_{\text{end}}$ in **App. F.3 (Fig. 7)**, under the multi-constraint multi-budget setting across all 26 tasks.
>
> As shown in the figure, there is a clear trend: **as $\beta_{\text{end}}$ increases from 0.5 to 0.9, both the average reward performance and cost performance of the policy show an increase trend, but the increase of cost remains within 0.1**. This indicates that when $\beta_{\text{end}}$ is smaller, the policy emphasizes safety satisfaction more strongly; whereas when $\beta_{\text{end}}$ is larger, the policy places more emphasis on reward maximization.
>
> Since $\beta_{\text{end}}$ is only applied during **evaluation**, this observation inspired us to explore an **Oracle setting**, where $\beta_{\text{end}}$ is slightly tuned for each task based on actual test-time performance. The results under this setting are shown in **Tab. 5 of App. F.3**. We observe that such post-hoc tuning during evaluation can further improve the overall safety performance of the policy.
>
> ### Q4 Extending CoPDT to online setting
>
> While we focus on the offline safe RL setting in this work, we agree that extending CoPDT to online scenarios is a highly promising direction for future research. In **Sec. 4.4 (Fig. 4)** and **App. F.7 (Fig. 10–12)**, we have demonstrated that CoPDT can perform efficient adaptation when provided with a small number (e.g., 10) of expert trajectories, indicating its potential for low-data transfer.
>
> However, in fully online settings, directly applying supervised learning-based approaches such as Online Decision Transformer (ODT) [4] may lead to poor learning efficiency, primarily due to the difficulty of exploring trajectories that satisfy safety constraints. This poses unique challenges in safe exploration. To address this, new algorithmic designs will be required—for example, determining how to choose the initial CTG and RTG values during exploration, how to sample them dynamically, and how to update the policy under safety-aware objectives.
>
> **We consider this an important and challenging direction, and leave it as a key avenue for future work.**
>
> ### Q5 Zero-shot generalization when learned with more similar tasks
>
> Thanks for the valuable suggestion, **pre-training on a larger set of similar tasks can indeed effectively enhance CoPDT’s zero-shot generalization capabilities**.
>
> We conducted **an additional experiment where we trained policies under a larger number of velocity thresholds** (from 2 to 6) in the AntVelocity and HopperVelocity environments. Specifically, in the 2-threshold setting, the thresholds were [2.57,2.62] for Ant and [0.37,0.74] for Hopper. In the 6-threshold setting, we used [2.42,2.47,2.55,2.57,2.62,2.67] for Ant and [0.37,0.42,0.52,0.62,0.67,0.74] for Hopper.
>
> For evaluation, beyond the original zero-shot generalization tasks (**AntV2** with threshold 2.52 and **HopperV2** with threshold 0.56), we further designed two additional zero-shot generalization tasks: **AntV7** (threshold 2.4) and **HopperV7** (threshold 0.35). Notably, the V2 tasks correspond to **interpolation** generalization settings, while the V7 tasks represent **extrapolation** settings. All results were obtained under a budget of 10, and evaluated using 3 random seeds.
>
> The experimental results are shown in the table below. It can be observed that when CoPDT is pre-trained on a larger number of similar tasks—similar to the setup in meta-RL—the model exhibits **significant improvements in safety performance** on both interpolation (V2) and extrapolation (V7) generalization tasks. The improvement is particularly notable in the Hopper environment, which may be attributed to the large threshold gap between the original two Hopper tasks, making pre-training on only two tasks insufficient. These results support the claim that our **constraint-prioritized prompt encoder not only captures task-specific memory but also possesses a certain degree of generalization capability to unseen constraints**.
>
> |  | CoPDT (2 similar tasks) | CoPDT (6 similar tasks) |
> | --- | --- | --- |
> |  | r↑  c↓ | r↑  c↓ |
> | AntV2 | 0.99, 2.18 | 0.98, 1.56 |
> | AntV7 | 0.99, 3.28 | 0.96, 2.34 |
> | HopperV2 | 0.52, 9.96 | 0.48, 0.04 |
> | HopperV7 | 0.24, 21.68 | 0.43, 4.57 |
> | Average | 0.69, 9.28 | 0.71, 2.13 |
>
> [1] Yu W, Gileadi N, Fu C, et al. Language to rewards for robotic skill synthesis[J]. arXiv preprint arXiv:2306.08647, 2023.
>
> [2] Xie T, Zhao S, Wu C H, et al. Text2reward: Reward shaping with language models for reinforcement learning[J]. arXiv preprint arXiv:2309.11489, 2023.
>
> [3] Li Z, Hu C, Wang Y, et al. Safe reinforcement learning with dual robustness[J]. IEEE Transactions on Pattern Analysis and Machine Intelligence, 2024.
>
> [4] Zheng Q, Zhang A, Grover A. Online decision transformer[C]//International Conference on Machine Learning. PMLR, 2022: 27042-27059.

---

> > ### Comment · Reviewer_YyH8 · 2025-08-04
> >
> > Thank you very much for your careful revisions and detailed responses to my comments. Most of my previous questions have been addressed or clarified, and I appreciate the effort you’ve dedicated to improving the manuscript. I would still recommend that you revisit and revise the figures and tables in the manuscript to enhance their clarity.

---

> > > ### Author Response · Authors · 2025-08-04
> > >
> > > Thank you very much for your response and valuable suggestions. We will revise the figures and tables accordingly in the next version of our paper to improve the clarity.

---

### Official Review · Reviewer_2Z9j · 2025-07-03

**Clarity:** 2
**Significance:** 3
**Originality:** 3
**Rating:** 4
**Confidence:** 3

**Summary:**

This paper proposes CoPDT (Constraint Prioritized Decision Transformer), a method for offline safe reinforcement learning in multi-task settings with varying constraints and budgets. CoPDT introduces two key components:
- A constraint prioritized prompt encoder that distinguishes between tasks.
- A CPRTG mechanism that dynamically generates Return-To-Go values conditioned on safety budgets, resolving conflicts between reward and safety objectives.

CoPDT outperforms several baselines under different settings of the OSRL benchmark.

**Questions:**

- Do you train CoPDT on a mix of PointButton1 and PointButton2 datasets, or across all Point robot datasets? Or does each environment have multiple cost functions? If it’s the latter, where do these multiple cost signals come from? Simply modifying the constraint threshold (e.g., a velocity limit that triggers a binary cost) may not justify calling it a separate constraint. Multiple costs should correspond to qualitatively different constraints. For a true multitask setup, training on a mix of datasets with different task semantics (e.g., PointButton and PointCircle) would be more compelling. This needs clarification.
- How are safe vs unsafe trajectory segments defined? Is the model trying to drive costs to zero, or just keep them under the given budget?
- If the expert trajectory used for prompt identification has zero cost, how does it convey information about the constraint? Or do expert demonstrations consume the full budget?
- How is the constraint type identified from the trajectory itself? If the constraint is not directly observable (like a rule about staying in a lane), how does the model infer its presence from data?
- Is Equation 9 used to train the CPRTG model q_phi? How exactly is this done, and how is Gt (the expert indicator) defined in practice? What counts as an expert trajectory?
- Comparing to FISOR in the multi-budget setting is problematic, as FISOR is not budget-aware and its performance is mostly fixed unless re-tuned.

**Ethical Concerns:**

["NO or VERY MINOR ethics concerns only"]

**Final Justification:**

The authors have addressed my concerns, which has given me a better understanding of the paper’s merits. As a result, I have increased my score.

**Limitations:**

Yes, the authors adequately address limitations and societal impact considerations. They explicitly acknowledge:
- The limitations of relying on binary cost signals.
- The difficulty of scaling to broader task distributions.
- The challenge of extrapolation errors in offline RL, mitigated in part by the CPRTG design.

**Paper Formatting Concerns:**

No concerns.

**Quality:**

3

**Strengths And Weaknesses:**

Strengths:
- The work addresses an important challenge in safe RL: learning from offline data and generalizing to multiple constraints and budgets.
- The CPRTG mechanism is a novel and well-motivated contribution that replaces manual RTG conditioning with a learned, CTG-conditioned alternative.
- The paper presents strong empirical results across a variety of evaluation setups.
- It includes extensive ablation studies and visualizations.

Weaknesses:
- The definition of multitask or multi-constraint settings is unclear. If the tasks only differ by simple modifications like velocity thresholds, it’s questionable whether they represent genuinely distinct or meaningful benchmarks. A more diverse or semantically varied tasks/constraints are required.
- The architecture includes many components: multiple encoders, decoders, and learned modules. This increases complexity and raises concerns about over-engineering.
- The comparison to CPQ is limited in value for the Single-constraint Multi-budget setting; BC-Safe (used in the oracle ablation) or more recent methods (below) show that CPQ is not a strong baseline.
  - OASIS [arXiv:2407.14653]
  - Latent Safety-Constrained Policies [arXiv:2412.08794]
  - FAWAC [arXiv:2412.08880]
  - Trajectory Classification for Safe RL [arXiv:2412.15429]
  - Constraint-Adaptive Policy Switching [arXiv:2412.18946], referenced in the introduction, notably supports multi-budget generalization.

---

> ### Author Rebuttal · Authors · 2025-07-31
>
> Thank you for your careful reviews and constructive suggestions. We appreciate your feedback and hope that our response addresses your concerns.
>
> ### Q1 Clarification of multi-constraint settings
>
> Thank you for your valuable comments. We would like to clarify that **our multi-constraint setting does not refer to handling multiple constraints within a single environment (e.g., PointButton1)**. Instead, **as described in Section 4.1, we train a single DT policy using a unified dataset that spans 26 diverse tasks**, including **PointButton1 through AntVelocity-v1**. This universal policy is then evaluated across all these environments and further tested in few-shot transfer scenarios to previously unseen tasks.
>
> Due to the heterogeneity in state and action spaces among the 26 tasks—for example, the state dimension of PointButton is 76, while that of CarCircle is 40—we employ task-specific encoders and decoders. These components help standardize the input and output dimensions for the shared Transformer core, allowing it to effectively operate across diverse environments.
>
> We sincerely apologize for the earlier lack of clarity in our explanation, and we truly appreciate your feedback. We will revise the manuscript to provide a more precise and detailed description of the multi-task setting to avoid any future confusion.
>
> ### Q2 Definition of safe and unsafe
>
> We thank the reviewer for the opportunity to clarify our problem formulation. Our work is situated within the **soft constraint formulation of safe RL**, a prominent and widely-adopted paradigm in the field [1, 2]. To be more precise, in this formulation, a trajectory is considered 'safe' if its total accumulated cost does not exceed a predefined budget. Therefore, the core objective of our model is to learn a policy that generates trajectories compliant with these specified cost constraints.
>
> ### Q3 Zero-cost expert trajectory
>
> CoPDT, is designed to be robust and **does not necessitate unsafe examples in the expert data to function**. In the specific scenario where an expert trajectory has zero cost, our method gracefully handles it as follows:
>
> - The entire trajectory is effectively considered a single, continuous 'safe patch'.
> - This 'safe patch' is then processed exclusively by the safe encoder $p_s$ to generate its representation.
> - Since no cost violations exist, the unsafe encoder $p_u$ is naturally bypassed and does not contribute to the final embedding.
>
> Crucially, our formulation in **Eqn. (5)** is designed to unify these cases, allowing the final representation to be computed seamlessly. We will clarify this point in the revised version to avoid possible misunderstandings.
>
> ### Q4 When constraint is not directly observable
>
> Typically, we assume that cost labels are directly available in offline datasets. However, in many real-world scenarios, we often encounter situations where only rule-based descriptions of costs are provided, without explicit cost labels or with only ambiguous or noisy cost labels.
>
> Under this setting, we believe a promising approach is to **apply the commonly used language-guided reward generation methods [3,4] to cost modeling, and integrate them with CoPDT**. Specifically, cost-related rule descriptions, along with observation and other relevant information, can be input into a LLM. The LLM can then generate a cost function that aligns with the provided rule. After cost function generation, its correctness can be verified and refined through human inspection or a small number of labeled samples. Then, this cost function can subsequently be used to annotate the offline dataset without cost labels. We have conducted preliminary experiments along this line, as discussed in our response to **Q2 of Reviewer YyH8**.
>
> ### Q5 Meaning of Eqn. (9)?
>
> Thank you for pointing this out. We would like to clarify that **Eqn. (9) is not used to train $q_\phi$**—rather, it is trained using Eqn. (8). **Eqn. (9) defines the target we aim to sample from $q_\phi$**. In this formulation, $G_t$ serves as an expert indicator. Under our setup, among trajectories that satisfy the CTG budget constraint, those **with higher returns are considered more "expert”**. Therefore, $p(\hat{R}_t \mid G_t, \hat{C}_t, s_t)$ represents the desired sampling distribution over $\hat{R}_t$ conditioned on $\hat{C}_t$ and $s_t$, such that the sampled $\hat{R}_t$ is more likely to correspond to trajectories aligned with the expert indicator $G_t$. Eqn. (9) then applies Bayes’ rule to decompose this distribution into two components: The first term reflects the likelihood under $q\_\phi$, i.e., the probability is higher when the sample is closer to the mean of $q\_\phi$; The second term, $p(G_t \mid \hat{C}_t, \hat{R}_t)$, denotes the likelihood that a trajectory with the given CTG and RTG is considered an expert. When CTG is fixed, trajectories with higher RTG are more likely to satisfy $G_t$ (importantly, $G_t$ is not used to directly label trajectories—it is an interpretable latent variable). The takeaway from Eqn. (9) is that, during sampling from $q\_\phi$, the objective is **not only to sample high-probability RTGs but also to prefer relatively larger RTGs that are more likely to satisfy the expert indicator**. We will improve the clarity in the next version.
>
> ### Q6 Comparison with FISOR?
>
> While FISOR is not inherently budget-aware, we include it as a strong baseline due to its SOTA performance in safety. FISOR is designed for hard constraint settings and does not accept a budget as input, we **conducted 12 separate runs for each task and grouped them into four categories, using $[10, 20, 40, 80]$ to normalize the cost accordingly**. From the safety perspective, this setup gives FISOR an advantage during evaluation, as the normalization scale of cost is relaxed for it. Despite this, CoPDT achieves comparable safety performance to FISOR, demonstrating that CoPDT maintains strong safety properties. We acknowledge that this comparison may not be perfectly aligned in terms of problem assumptions, but we believe it still provides useful insights into safety performance across settings.
>
> ## Q7 New baselines
>
> Thank you very much for your valuable suggestion. We have additionally provided experimental results comparing CoPDT with **OASIS [5] and LSPC [6]** in the **Single-constraint Multi-budget setting, and the conclusions remain consistent with our previous findings**. Due to time constraints, these experiments were conducted on the first 16 tasks. Since the official OASIS code does not include hyperparameter settings for the SafetyGymnasium tasks we tested, we followed a similar practice to that used in their BulletGym experiments and adopted the hand-set (RTG, CTG) pairs provided in the OSRL codebase as generation-condition hyperparameters for OASIS.
>
> The results are shown in the table below. OASIS, which leverages diffusion models for data generation, achieved promising results in the BulletGym environment. However, due to time constraints, we were unable to perform extensive hyperparameter tuning for OASIS in the SafetyGymnasium environments, which may have limited its performance and failed to fully showcase its potential. On the other hand, LSPC—a hard-constraint baseline similar to FISOR—demonstrates strong safety performance and achieves constraint satisfaction in most environments. Nevertheless, **CoPDT still outperforms LSPC in terms of overall safety, underscoring the effectiveness of the CPRTG mechanism in balancing reward and safety under the multi-budget setting**, in line with our prior conclusions. We will include the corresponding experimental results in the next version.
>
> | Task | OASIS | LSPC | CoPDT |
> | --- | --- | --- | --- |
> |  | r↑  c↓ | r↑  c↓ | r↑  c↓ |
> | PointButton1 | **0.00, 0.97** | 0.16, 1.90 | **0.05, 0.66** |
> | PointButton2 | -0.16, 1.12 | 0.17, 1.70 | 0.14, 1.41 |
> | PointCircle1 | 0.59, 5.11 | 0.55, 6.64 | **0.50, 0.63** |
> | PointCircle2 | 0.69, 7.47 | 0.62, 5.31 | **0.61, 0.98** |
> | PointGoal1 | **0.41, 0.97** | **0.25, 0.27** | **0.36, 0.56** |
> | PointGoal2 | 0.37, 1.71 | **0.25, 0.85** | 0.31, 1.02 |
> | PointPush1 | **0.00, 0.67** | **0.13, 0.97** | **0.19, 0.88** |
> | PointPush2 | **-0.72, 0.52** | **0.11, 0.89** | 0.19, 1.47 |
> | CarButton1 | 0.00, 9.29 | -0.07, 1.37 | **0.07, 0.74** |
> | CarButton2 | -0.62, 1.30 | -0.15, 1.52 | -0.02, 1.33 |
> | CarCircle1 | 0.53, 5.26 | 0.45, 3.50 | 0.51, 3.34 |
> | CarCircle2 | 0.61, 17.71 | **0.26, 0.00** | **0.28, 0.98** |
> | CarGoal1 | **-0.57, 0.18** | **0.24, 0.51** | **0.33, 0.47** |
> | CarGoal2 | 0.01, 3.50 | **0.12, 0.59** | **0.19, 0.81** |
> | CarPush1 | -0.02, 1.41 | **0.22, 0.86** | **0.20, 0.67** |
> | CarPush2 | -0.18, 1.21 | **0.09, 0.93** | **0.07, 0.73** |
> | Average | 0.06, 3.65 | 0.21, 1.74 | 0.24, 1.04 |
>
> [1] Garcıa J, Fernández F. A comprehensive survey on safe reinforcement learning[J]. Journal of Machine Learning Research, 2015, 16(1): 1437-1480.
>
> [2] Gu S, Yang L, Du Y, et al. A review of safe reinforcement learning: Methods, theories and applications[J]. IEEE Transactions on Pattern Analysis and Machine Intelligence, 2024.
>
> [3] Yu W, Gileadi N, Fu C, et al. Language to rewards for robotic skill synthesis[J]. arXiv preprint arXiv:2306.08647, 2023.
>
> [4] Xie T, Zhao S, Wu C H, et al. Text2reward: Reward shaping with language models for reinforcement learning[J]. arXiv preprint arXiv:2309.11489, 2023.
>
> [5] Yao Y, Cen Z, Ding W, et al. Oasis: Conditional distribution shaping for offline safe reinforcement learning[J]. Advances in Neural Information Processing Systems, 2024, 37: 78451-78478.
>
> [6] Koirala P, Jiang Z, Sarkar S, et al. Latent Safety-Constrained Policy Approach for Safe Offline Reinforcement Learning[C]//The Thirteenth International Conference on Learning Representations, 2025.

---

> > ### Comment · Reviewer_2Z9j · 2025-08-05
> >
> > Thank you to the authors for their response. Overall, the rebuttal is satisfactory. However, the argument regarding FISOR remains unconvincing. Increasing the cost budget does not improve FISOR’s actual performance, its cumulative reward remains unchanged. While the normalized cost may decrease, the point of increasing the budget is to evaluate whether a method can accumulate more reward, which FISOR fails to do. As such, it is not a suitable baseline for this purpose.
> >
> > That said, given the clarifications provided, I have updated my score and now recommend a borderline accept. I encourage the authors to incorporate the additional results and edits discussed during the rebuttal into the final manuscript.

---

> > > ### Author Response · Authors · 2025-08-05
> > >
> > > Thank you very much for your thoughtful feedback and detailed review. We appreciate your updated evaluation and constructive comments, and will make sure to incorporate the additional results and edits during the rubuttal into the next revision of our paper.

---

### Official Review · Reviewer_9wTp · 2025-07-03

**Clarity:** 3
**Significance:** 3
**Originality:** 3
**Rating:** 5
**Confidence:** 4

**Summary:**

The paper introduces CoPDT (Constraint Prioritized Decision Transformer), a novel framework for offline safe reinforcement learning (RL) in multi-task settings where multiple safety constraints and budgets vary across tasks. COPDT introduced two key components: a constraint prioritized prompt encoder that distinguishes safe and unsafe trajectory segments to identify task-specific constraints, and a constraint prioritized returns-to-go generator that dynamically produces conflict-free RTG values conditioned on safety budgets and contextual signals. These allow CoPDT to generate adaptive, safe policies using only offline data, without requiring environment interaction. Extensive experiments demonstrate that CoPDT outperforms state-of-the-art baselines in both safety compliance and generalization, particularly excelling in multi-budget and multi-constraint scenarios.

**Questions:**

Questions echo the weakness,
1. Choosing $\beta_t$ relies on the structure that $q_\phi(R_t|C_t)$ is Gaussian, which is a strong assumption. What if the underlying distribution is not Gaussian?
2. Any examples or theoritical intuitions of defining the causal structure $p(R_t|C_t)$ in line 163?

**Ethical Concerns:**

["NO or VERY MINOR ethics concerns only"]

**Final Justification:**

I have read the comments from other reviewers and AC. I will keep my score.

**Limitations:**

Yes.

**Quality:**

3

**Strengths And Weaknesses:**

## Strength
1. The writing is clear.
2. The motivation and background of safe multi-task RL are well explained.
3. The problems described in the paper are adequate and handled properly.
4. The separation of safe and unsafe encoders looks novel to me.
5. The causal structure of CTG and RTG, and introducing the variable $G_t$ indicating the optimality, looks novel to me, which might create novel viewpoints in balancing the returns and safety in the community of safe RL.
6. The experimental results are extensive and convincing.

## Weakness
1. The beta-decay design seems artificial, although it makes sense.
2. The notation system does not highlight the multi-task nature of the problem formulation, which should be the core contribution of the paper. The authors should revise the notations.
3. The causal structure, despite its novelty, lacks some intuitive explanations and examples. It will be good to include some examples.

---

> ### Author Rebuttal · Authors · 2025-07-31
>
> Thank you for your kind and thoughtful reviews. We appreciate your feedback and hope that our response addresses your concerns.
>
> ### Q1 Choosing $\beta_t$ when the underlying distribution is not Gaussian?
>
> In our current implementation, we model $q_\phi$ as a Gaussian distribution, as it is generally sufficient to handle most practical scenarios. However, we acknowledge that this assumption may not hold in more complex scenarios. In such cases, we believe that **leveraging the strong multi-modal distribution modeling capabilities of generative models—such as diffusion models or flow matching—provides a more suitable alternative** for modeling $q_\phi$. Under this setup, the choice of $\beta_t$ can remain consistent with the Gaussian case. Specifically, to sample $\hat{R}_t$, we first draw $X$ samples from a standard normal distribution. These samples are then transformed using a diffusion model or flow matching to generate $X$ candidate values. Finally, we select the $\beta_t$-quantile among these candidates as the final $\hat{R}_t$.
>
> ### Q2 Intuitive explanations about modeling RTG on CTG?
>
> **Intuitively:** To illustrate the motivation behind our design, consider a simple example. In our framework, the CTG represents the goal of safety satisfaction, while the RTG corresponds to reward maximization. The DT policy acts as a **goal-conditioned policy**, where both CTG and RTG jointly specify the intended behavior.
>
> Suppose an agent operates along a 1D axis and starts at the origin. The CTG defines a safety region—e.g., $X \in [-10, 10]$—within which the agent must remain. Meanwhile, the reward function incentivizes movement in the positive X direction, so the RTG effectively specifies a target position. If the RTG is set to 20, the agent will attempt to reach $X = 20$, which lies outside the safe region, resulting in a conflict between the safety and reward objectives. In this case, the agent may violate safety constraints in pursuit of the reward.
>
> To avoid such conflicts, our **CPRTG mechanism** retrieves feasible (CTG, RTG) pairs from the offline dataset. For instance, for a given CTG, the dataset might contain trajecotries with RTG values of 5, 8, 9, and 9.8. **Since these trajectories already satisfy the safety constraints, their associated RTGs are guaranteed to be compatible with the CTG**. Among these candidates, we select the one with the **highest return** (e.g., 9.8), thereby maximizing reward **while ensuring safety**.
>
> **Formally:** More detailly, modeling RTG using $q_\phi$ conditioned on CTG can be interpreted from two complementary perspectives: **(1) Resolving the conflict between RTG and CTG in multi-budget safe RL; (2) Reducing extrapolation error from an offline RL perspective.**
>
> For the first point, we have visualized the conflict between RTG (representing reward maximization) and CTG (representing adherence to safety constraints) in **Fig. 2(b) and 2(c) of Sec. 4.3**. The plots show that, for a DT model, when CTG is held fixed and RTG is increased, both the reward and cost performance of the resulting policy increase accordingly. In contrast, when RTG is fixed and CTG is varied, the performance remains largely unchanged. This asymmetry indicates that the **DT model tends to prioritize RTG over CTG, thereby potentially neglecting safety requirements**. To address this issue in safe RL—where satisfying safety constraints is often a primary concern—we propose using $q_\phi$ to automatically generate RTG values conditioned on a given CTG. These RTG values are learned from offline data and are generally **more conservative than externally specified RTG targets**, which helps mitigate the conflict between reward and safety objectives.
>
> From the offline RL perspective, DT learns policies through behavior cloning via supervised learning. Hence, its performance heavily relies on how well the **input distribution aligns with that of the offline dataset**. If CTG is regarded as an externally specified safety target and **RTG is treated as part of the input state**, minimizing the extrapolation error requires aligning the policy input distribution with that of the offline data. To achieve this, we model RTG using $q_\phi$ learned from the offline dataset. This interpretation is further elaborated in **App. A**.
>
> ### Q3 The notation system does not highlight the multi-task nature
>
> Thank you for the constructive suggestion. We will revise our notation system accordingly in the next version. Specifically, we will explicitly incorporate the task embedding into the formulations in Sec. 3.2 and 3.3 to better highlight the multi-task nature of our approach.

---

> > ### Comment · Reviewer_9wTp · 2025-08-04
> >
> > Thanks for the rebuttal and for taking my suggestions into consideration. All my concern is addressed. Please consider adding the examples in the revised manuscript.

---

> > > ### Author Response · Authors · 2025-08-04
> > >
> > > Thank you very much for your response. We will incorporate the examples in the next revision of our paper.

---

### Decision · Program_Chairs · 2025-09-17

**Decision:**

Accept (poster)

**Comment:**

This paper proposes encoding cost constraints to inform value prediction, which is a novel and well-motivated idea. The approach is clearly presented, and the experimental results show meaningful benefits across the evaluated settings. The reviewers were generally positively inclined toward the paper, recognizing both the conceptual contribution and the empirical evidence.

Some concerns were raised during the review process, including questions about the scope of the evaluation, the generality of the approach, and whether the novelty lies primarily in combining existing ideas. The authors provided clear and thoughtful responses to these points, which helped to address the main issues raised. Incorporating these clarifications into the final version will further strengthen the paper and make the contribution more evident.